# Adaptive Conformal Guidance for Learning under Uncertainty

**Rui Liu[1], Peng Gao[2], Yu Shen[3], Ming Lin[1], Pratap Tokekar[1]**
[1]University of Maryland, College Park
[2]North Carolina Satate University
[3]Adobe Research

## Abstract

Learning with guidance has proven effective across a wide range of machine learning systems. Guidance may, for example, come from annotated datasets in supervised learning, pseudo-labels in semi-supervised learning, and expert demonstration policies in reinforcement learning. However, guidance signals can be noisy due to domain shifts and limited data availability and may not generalize well. Blindly trusting such signals when they are noisy, incomplete, or misaligned with the target domain can lead to degraded performance. To address these challenges, we propose Adaptive Conformal Guidance (`AdaConG`), a simple yet effective approach that dynamically modulates the influence of guidance signals based on their associated uncertainty, quantified via split conformal prediction (CP). By adaptively adjusting to guidance uncertainty, `AdaConG` enables models to reduce reliance on potentially misleading signals and enhance learning performance. We validate `AdaConG` across diverse tasks, including knowledge distillation, semi-supervised image classification, gridworld navigation, and autonomous driving. Experimental results demonstrate that `AdaConG` improves performance and robustness under imperfect guidance, e.g., in gridworld navigation, it accelerates convergence and achieves over $6\times$ higher rewards than the best-performing baseline. These results highlight `AdaConG` as a broadly applicable solution for learning under uncertainty.

## 1 Introduction

Machine learning systems often rely on some form of guidance during training to enhance performance (Hinton, 2015; Romero et al., 2014; Zagoruyko and Komodakis, 2016; Passalis and Tefas, 2018), bootstrap learning in data-scarce scenarios (Sohn et al., 2020; Zhang et al., 2021), and improve sample efficiency (Hu et al., 2023; Bhaskar et al., 2024b). While such guidance has proven valuable, a critical challenge arises when this guidance is noisy. In supervised learning, richly annotated datasets provide guidance to enhance model performance, and leveraging pretrained models (Hinton, 2015; Romero et al., 2014; Zagoruyko and Komodakis, 2016; Passalis and Tefas, 2018; Kim et al., 2018; Wang et al., 2023; Xue et al., 2022; Huo et al., 2024; Gu et al., 2023; Jin et al., 2023) has become an effective strategy to boost performance and enable deployment in resource-constrained environments, with lighter-weight models that either use reduced modalities or smaller architectures during inference (Shen et al., 2023; Gu et al., 2023; Liu et al., 2025b;c). These teacher–student frameworks allow the student to benefit from the teacher's superior predictions. However, this setup critically assumes that the teacher's outputs remain reliable when applied to the student's target domain. In practice, domain shifts can render the teacher's guidance noisy or misleading.

Similarly, semi-supervised learning (SSL) expands the effective training set through pseudo-labeling to bootstrap learning in data-scarce scenarios (Sohn et al., 2020; Zhang et al., 2021). However, the quality of these pseudo-labels may not be high due to the inherent uncertainty. This uncertainty stems from several factors (Scherer et al., 2022; Xia et al., 2023; Kage et al., 2024): the small labeled set may not fully represent the data distribution, the model's early mistakes can propagate through self-training, and the confidence thresholds for pseudo-labeling may not perfectly filter out incorrect labels. As a result, noisy pseudo-labels can misguide the learning process, potentially reinforcing errors and degrading model performance.

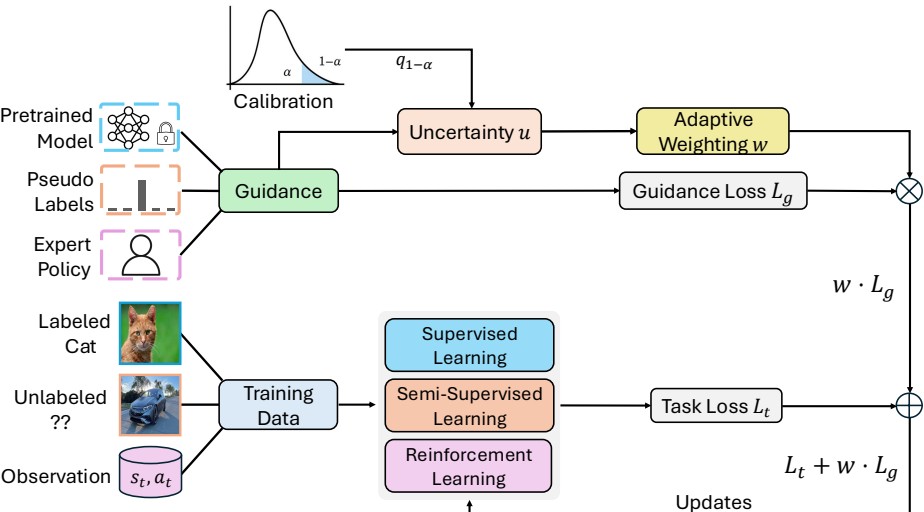

Figure 1: **Overview of the `AdaConG` approach**. `AdaConG` leverages split CP with calibration to quantify the uncertainty of guidance signals and adaptively modulate their influence. The estimated uncertainty $u$ is converted into an adaptive weight $w$, which reweights the guidance loss. This weighted guidance loss is then combined with the task loss to update the model, enabling effective learning under uncertain guidance.

In reinforcement learning, agents employ imitation-learned policies for guidance (Hu et al., 2023; Bhaskar et al., 2024b) to reduce exploration demands and improve sample efficiency, yet a critical challenge emerges when the target environment differs from the ones used for the expert demonstrations. While imitation learning provides valuable behavioral priors, these policies often struggle to generalize beyond their training distribution. When the RL agent encounters states or observations outside the expert's demonstration space, the imitation policy's guidance becomes increasingly noisy, potentially leading to suboptimal exploration.

In all these scenarios, blindly relying on noisy guidance can propagate errors and lead to suboptimal model performance, as the learning system overfits to potentially misleading information, yet discarding potentially valuable guidance wastes computational resources and domain knowledge. Despite its fundamental importance, the challenge of effectively leveraging noisy guidance while maintaining robust learning capabilities remains largely unaddressed across machine learning systems. The central question becomes: *How can we effectively leverage potentially valuable guidance while appropriately accounting for its uncertainty to ensure robust model learning?*

While prior works have explored uncertainty-aware learning (Angelopoulos et al., 2020; Mossina et al., 2024; Lu et al., 2022; Karimi and Samavi, 2023; Zhao et al., 2024; Su et al., 2025; Zhang et al., 2024; Edupuganti et al., 2020; Kwon et al., 2020; Wang et al., 2020; Gao and Zhang, 2021; Gao et al., 2023; Liu et al., 2023; 2024b; 2025e; 2026), they typically focus on heuristic uncertainty estimates, domain-specific solutions, or post-hoc calibration. Monte Carlo dropout (Zhao et al., 2024), entropy-based reweighting (Su et al., 2025; Liu et al., 2025d), or perturbation of the distillation loss Zhang et al. (2024), tackle uncertainty by modulating loss terms via heuristics or series approximations. However, these approaches remain confined to supervised classification, and the effectiveness may degrade under distribution shifts. Other methods address uncertainty only within narrow domains, e.g., medical imaging (Edupuganti et al., 2020; Kwon et al., 2020; Wang et al., 2020) or human-robot interaction Gao and Zhang (2021); Gao et al. (2023). Techniques such as conformal prediction (CP) have been used primarily to calibrate model outputs after training (Angelopoulos et al., 2020; Mossina et al., 2024; Lu et al., 2022; Karimi and Samavi, 2023).

In contrast, we propose the first framework `AdaConG` that embeds split CP directly into the training loop to adaptively weight guidance across supervised, semi-supervised, and reinforcement learning settings. Simple yet effective, our method serves as a broadly applicable solution for incorporating uncertainty-aware guidance, as illustrated in Fig. 1. While CP has primarily been applied for post-hoc calibration, its potential to inform real-time training dynamics remains underexplored. CP provides a distribution-free, model-agnostic approach to constructing prediction sets (Shafer and Vovk, 2008; Angelopoulos and Bates, 2021), making it suited across diverse learning systems. Un-

like heuristic uncertainty estimates such as entropy (Namdari and Li, 2019) or maximum softmax probability (MSP) (Pearce et al., 2021) which rely on the softmax outputs that are often overconfident and poorly calibrated, CP provides more rigorous uncertainty estimates (Shafer and Vovk, 2008; Angelopoulos and Bates, 2021), even when the underlying distribution changes (Zhou et al., 2025; Gibbs and Candès, 2021).

We validate our approach through extensive experiments across multiple tasks, including knowledge distillation, semi-supervised image classification, gridworld navigation, and autonomous driving, demonstrating improvements in performance and robustness compared to conventional methods. Our results underscore the critical importance of adaptive uncertainty weighting in scenarios where guidance signals may be imperfect, providing a solution toward more reliable machine learning systems. Overall, the key contributions of our work are as follows:

- We propose `AdaConG`, an approach that adaptively modulates the influence of guidance signals based on their uncertainty, ensuring effective learning without over-relying on unreliable guidance.
- `AdaConG` is broadly applicable across diverse learning systems including supervised, semi-supervised, and imitation-guided reinforcement learning.
- `AdaConG` can extract useful insights even when guidance underperforms, unlike conventional methods that assume guidance is always trustworthy. In gridworld navigation, it enables faster convergence and achieves over $6\times$ higher rewards than the strongest baseline.

## 2 RELATED WORK

**Learning with Guidance.** Learning with guidance has been a common and effective strategy across various machine learning systems. In supervised learning, annotated datasets provide explicit guidance for model training, and many works leverage pretrained models to further boost performance. For instance, Hinton (2015); Jin et al. (2023); Sun et al. (2024) focused on transferring soft probabilities from teacher models' logits to guide student models, while Romero et al. (2014); Zagoruyko and Komodakis (2016); Passalis and Tefas (2018); Kim et al. (2018) emphasized transferring intermediate features. Cross-modal guidance has also been explored: Wang et al. (2023) proposed a prototype-based distillation method for medical image segmentation, where a multi-modal teacher guides a single-modal student; Shen et al. (2023) introduced the Auxiliary Modality Learning (AML) framework, enabling a teacher model with access to multiple modalities to transfer knowledge to a student operating with fewer modalities at test time; and Liu et al. (2025b) extended this idea to multi-agent settings. In semi-supervised learning (Sohn et al., 2020; Zhang et al., 2021), pseudo-labels generated from unlabeled data provide implicit guidance to bootstrap learning with limited labeled data. In reinforcement learning, pretrained imitation learning (IL) (Bhaskar et al., 2024a; Liu et al., 2024a; 2025a) policies derived from expert demonstrations have been used to guide RL agents and improve sample efficiency (Hu et al., 2023; Bhaskar et al., 2024b).

However, a key limitation of most existing methods is their reliance on static guidance, which assumes that guidance signals are always reliable. This assumption often breaks down when guidance contains uncertainty, due to domain shifts in supervised learning, limited labeled data in semi-supervised learning, or generalization constraints of IL policies in reinforcement learning. In contrast, `AdaConG` introduces a principled approach that dynamically modulates the influence of guidance signals based on their associated uncertainty, offering a simple yet effective, and broadly applicable solution for incorporating uncertainty-aware guidance.

**Conformal Prediction.** Conformal prediction (CP) (Angelopoulos et al., 2020; Angelopoulos and Bates, 2021; Mossina et al., 2024; Karimi and Samavi, 2023; Tibshirani et al., 2019; Shafer and Vovk, 2008; Vovk et al., 2020) is a non-parametric, distribution-free, and model-agnostic framework designed to provide reliable prediction sets. In machine learning systems, CP has primarily been utilized for post-hoc uncertainty calibration. For instance, Angelopoulos et al. (2020) introduced an algorithm that adapts any image classifier to output predictive sets containing the true label with a user-specified probability. Mossina et al. (2024) proposed a computationally lightweight approach to quantify predictive uncertainty in semantic image segmentation using CP. Similarly, Lu et al. (2022) applied CP to deep learning models for grading the severity of spinal stenosis in lumbar spine MRI, while Karimi and Samavi (2023) leveraged CP to measure uncertainty in deep learning models.

Despite recent advancements, the application of CP to inform real-time training dynamics remains underexplored. In this work, we extend split CP to learning with guidance under uncertainty, using it

as a module for adaptive weighting. By modulating the uncertainty of the guidance signal, we enable the model to reduce dependence on potentially misleading guidance and encourages the model to discover patterns that may be overlooked when strictly following uncertain guidance.

## 3 APPROACH

**Preliminaries.** In our framework, we leverage split conformal prediction (CP) to conformalize a guidance signal and quantify its uncertainty. CP is a distribution-free method that provides prediction sets with guaranteed coverage levels, regardless of the underlying model or data distribution (Angelopoulos and Bates, 2021; Shafer and Vovk, 2008).

Split CP uses a nonconformity score $s'$ to measure how unusual a prediction is for a new test input, based on a calibration set $\mathcal{D}_{\text{cal}}$, a held-out dataset used to compute the empirical distribution of nonconformity scores. The score $s'$ can be defined in various ways. For instance, in regression, it is often the absolute residual $s' = |\bar{y} - \hat{y}(\bar{x})|$, where $\bar{y}$ is the ground truth and $\hat{y}(\bar{x})$ is the model's prediction for an input $\bar{x} \in \mathcal{D}_{\text{cal}}$. In classification, a common choice is the confidence score $s' = 1 - p_{\bar{y}}(\bar{x})$, where $p_{\bar{y}}(\bar{x})$ is the model's estimated probability for the true class $\bar{y}$. Additional examples can be found in (Angelopoulos and Bates, 2021; Shafer and Vovk, 2008). Given a calibration set, we compute the quantile $q_{1-\alpha}$ of the nonconformity scores with $\alpha$ as an allowable error rate. The quantile is denoted as $q_{1-\alpha} = \text{Quantile}_{1-\alpha}(s'_1, s'_2, \ldots, s'_{|\mathcal{D}_{\text{cal}}|})$, representing a threshold below which $1 - \alpha$ of the data falls. This threshold is then used to construct prediction sets. For a test input $x_{\text{test}}$, the prediction set is constructed as $\mathcal{C}(x_{\text{test}}) = \{y : s'(x_{\text{test}}, y) \leq q_{1-\alpha}\}$. Under the assumption of exchangeability, the coverage guarantee holds that the probability of the true label $y_{\text{test}}$ falling within $\mathcal{C}(x_{\text{test}})$ satisfies $P(y_{\text{test}} \in \mathcal{C}(x_{\text{test}})) \geq 1 - \alpha$ (Angelopoulos and Bates, 2021).

**Adaptive Conformal Guidance.** `AdaConG` is a general framework, as illustrated in Fig. 1. We show how to use it in supervised, semi-supervised, and imitation-guided reinforcement learning settings. The core idea is to learn adaptive weights based on the uncertainty of the guidance signal, enabling dynamic modulation of its influence during training.

SUPERVISED LEARNING. We consider a supervised learning problem in which a pretrained model guides the training of a target model under potential domain shift. Formally, let the source domain dataset be denoted as $\mathcal{D}_s$, and the (shifted) target domain dataset as $\mathcal{D}_t$. We represent the pretrained model on $\mathcal{D}_s$ as $f_p : \mathcal{X} \to \mathcal{Y}$, and the target model under training as $f_t : \mathcal{X} \to \mathcal{Y}$, where $\mathcal{X}$ is the input space and $\mathcal{Y}$ is the output space. Our goal is to leverage $f_p$ to bootstrap the learning of $f_t$ so that it outperforms both supervised training from scratch and standard knowledge distillation under domain shift. To do so, we introduce an adaptive weighting mechanism based on split CP to modulate the guidance of the pretrained model. This is particularly important when $\mathcal{D}_t$ differs from $\mathcal{D}_s$, making $f_p(x)$ uncertain for input $x \in \mathcal{D}_t$.

Given the target domain dataset $\mathcal{D}_t$, we split it into three subsets: the training set $\mathcal{D}_{\text{train}}^t$, used to train the target model $f_t$; the calibration set $\mathcal{D}_{\text{cal}}$, used to transform any heuristic measure of uncertainty from the pretrained model $f_p$ into a rigorous one; and the testing set $\mathcal{D}_{\text{test}}$ for validate the target model performance. This setup ensures that the calibration set is representative of the inputs on which the guidance will be applied. We conformalize the pretrained model $f_p$ following the approach as described in Section 3. To quantify the guidance uncertainty, we leverage the size of the prediction set $\mathcal{C}(x)$ for an input $x$. Specifically, we define the guidance uncertainty as

$$u(x) = g(|\mathcal{C}(x)|), \tag{1}$$

where $g$ is a mapping ensuring $u(x) \in [0, 1]$ (e.g., $g(n) = \frac{n-1}{K-1}$ for a $K$-class problem). The adaptive weight is computed as

$$w(x) = h(u(x)), \tag{2}$$

where $h$ is a monotonically decreasing function (e.g., exponential decay $h(u) = \exp(-\kappa u)$ with a temperature $\kappa > 0$) such that high uncertainty results in a lower weight. Then we define the loss function of training the target model as $\mathcal{L} = \lambda_{\text{task}}\mathcal{L}_t + w(x) \cdot \lambda_{\text{guide}}\mathcal{L}_g$, where $\mathcal{L}_t$ is the task loss (e.g., cross entropy loss), $\mathcal{L}_g$ is the guidance loss (e.g., KL divergence between target model and pretrained model logits), $\lambda_{\text{task}}$ and $\lambda_{\text{guide}}$ are the coefficients. Essentially, the adaptive weighting mechanism of `AdaConG` allows the model to balance between relying on the pretrained guidance and self-exploration.

SEMI-SUPERVISED LEARNING.  In semi-supervised learning (SSL), our goal is to leverage a limited set of labeled data $\mathcal{D}_l = \{(x_i, y_i)\}_{i=1}^{N_l}$ along with a large set of unlabeled data $\mathcal{D}_u = \{x_i\}_{i=1}^{N_u}$. For each unlabeled sample $x \in \mathcal{D}_u$, the model $f$ produces a prediction $\hat{y} = f(x_{\text{weak}})$ using a weakly augmented view $x_{\text{weak}}$. The pseudo-label $\tilde{y}$ is obtained by thresholding the model's output on this weakly augmented input. The same model is then trained to predict $\tilde{y}$ from a strongly augmented version $x_{\text{strong}}$ of the same input.

Along with obtaining pseudo-labels, we introduce an adaptive weighting mechanism grounded in split CP to modulate the influence of each pseudo-label during training. We construct a calibration set $\mathcal{D}_{\text{cal}}$ by taking the labeled data and applying the same weak augmentation used for the unlabeled data. The CP procedure for generating a prediction set $\mathcal{C}(x)$ for an unlabeled sample $x$ follows the approach detailed in Section 3. The adaptive weight $w(x)$ is defined similarly as in Eq. 2. Then the unsupervised loss is defined as $\mathcal{L}_u = \frac{1}{|\mathcal{D}_u|} \sum_{x \in \mathcal{D}_u} w(x) \, \ell\big(f(x_{\text{strong}}), \tilde{y}\big)$, where $\ell(\cdot, \cdot)$ denotes the cross-entropy loss commonly used in SSL, which enforces consistency regularization between the strongly augmented prediction and the pseudo-label. This loss is adaptively weighted by the confidence of the pseudo-label $w(x)$. The supervised loss over the labeled set is given by another cross-entropy loss $\mathcal{L}_s = \frac{1}{|\mathcal{D}_l|} \sum_{(x,y) \in \mathcal{D}_l} \ell\big(f(x), y\big)$. The final objective function, which balances the contributions from both supervised and unsupervised components, is defined as $\mathcal{L} = \mathcal{L}_s + \lambda_u \mathcal{L}_u$, with $\lambda_u$ controlling the relative weight of the unsupervised loss.

IMITATION-GUIDED REINFORCEMENT LEARNING.  Consider a Markov Decision Process defined by the tuple $\{\mathcal{S}, \mathcal{A}, \mathcal{P}, \mathcal{R}, \gamma\}$, where $\mathcal{S}$ is the state space, $\mathcal{A}$ is the action space, $\mathcal{P}$ is the transition dynamics, $\mathcal{R}$ is the reward function, and $\gamma$ is the discount factor. We focus on off-policy RL methods due to their higher sample efficiency. We focus on cases where there is an imitation policy learned from expert demonstrations to guide the reinforcement learning.

To quantify uncertainties of the IL and RL policy, we use the nonconformity score $s(s, a) = -\log \pi(a|s)$. For the static imitation policy $\pi_{\text{I}}$, we pre-collect a calibration set $\mathcal{D}_{\text{cal, I}} = \{(s_i, a_i)\}_{i=1}^{N}$ by rolling out $\pi_{\text{I}}$ in the target environment. Then we compute a constant quantile $\hat{q}_{\text{I}}$ for the IL policy. For the RL policy $\pi_{\text{R}}^{(t)}$, we leverage adaptive CP (Zhou et al., 2025; Gibbs and Candès, 2021), maintaining a dynamic calibration set $\mathcal{D}_{\text{cal, R}}^{(t)}$ via a sliding window of size $N$, which we initialize as $\mathcal{D}_{\text{cal, R}}^{(0)} = \mathcal{D}_{\text{cal, I}}$. At each subsequent training step $t$, we add a new batch of $m$ state-action pairs from rollouts of $\pi_{\text{R}}^{(t)}$ and discard the oldest. Then we update the RL quantile $\hat{q}_{\text{R}}^{(t)}$ using an exponential moving average (EMA): $\hat{q}_{\text{R}}^{(t)} \leftarrow (1 - \rho)\hat{q}_{\text{R}}^{(t-1)} + \rho\tilde{q}_{\text{R}}^{(t)}$, where $\tilde{q}_{\text{R}}^{(t)}$ is computed from the current window's scores and $\rho$ is a smoothing factor. We warm-start the RL quantile by initializing it with the quantile of the imitation policy, i.e., $\hat{q}_{\text{R}}^{(0)} = \hat{q}_{\text{I}}$. Finally, the IL and RL policy uncertainties are defined as $u_{\text{I}}(s) = g(|\mathcal{C}_{\text{I}}(s)|)$ and $u_{\text{R}}(s) = g(|\mathcal{C}_{\text{R}}(s)|)$, using the static quantile $\hat{q}_{\text{I}}$ for $\pi_{\text{I}}$ and the adaptive quantile $\hat{q}_{\text{R}}^{(t)}$ for $\pi_{\text{R}}^{(t)}$, where $g$ is a mapping, e.g., the identity function. The loss for training the RL policy is defined as $\mathcal{L} = \mathcal{L}_t + w(s) \cdot \mathcal{L}_g$, where $\mathcal{L}_t$ is the task loss (e.g., $\mathbb{E}\left[-\log \pi_{\text{R}}(a|s) \cdot A(s, a)\right]$ with $A(s, a)$ as the advantage function), which corresponds to the RL objective. $\mathcal{L}_g = \mathbb{E}\left[\text{KL}(\pi_{\text{R}}(\cdot|s) \, \| \, \pi_{\text{I}}(\cdot|s))\right]$ is the KL divergence between the RL policy and the IL policy, serving as the guidance loss. The weight $w$ is defined as: $w(s) = h(u_{\text{I}}(s), u_{\text{R}}(s))$, e.g., $w(s) = \frac{\exp(-u_{\text{I}}(s))}{\exp(-u_{\text{I}}(s)) + \exp(-u_{\text{R}}(s))}$. In RL, the CP set measures a policy's self-consistency, not action correctness, and we use $w(s)$ as an uncertainty-driven guidance weight. This reduces reliance on imitation guidance when the IL policy's uncertainty is high. The same adaptive weight $w(s)$ is also used during data collection: at each state, the agent selects $a_{\text{I}}$ with probability $w(s)$ and $a_{\text{R}}$ with probability $1 - w(s)$. This sampling mechanism is the practical realization of the adaptive guidance and complements the KL-based regularization in the objective.

## 4  EXPERIMENTS

To validate our approach, we conduct experiments across a diverse range of tasks, including knowledge distillation, semi-supervised image classification, gridworld navigation, and autonomous driving steer prediction.

**Knowledge Distillation.**  We first evaluate our framework's effectiveness in improving classification performance over traditional supervised learning by leveraging knowledge distillation from a pretrained teacher to a student model. This evaluation is conducted under domain shift and noise,

where the teacher may underperform. Specifically, we aim to address the following question: When the teacher model is underperformance, can it still provide useful "dark knowledge" to enhance the performance of the student model beyond what is achievable by training from scratch?

EXPERIMENTAL SETUP. We conduct our experiments on the CIFAR-100 dataset (Krizhevsky et al., 2009) and report the mean and standard deviation over four repeated runs. We introduce domain shifts to the datasets by adding Gaussian noise of zero mean and a standard deviation of 0.05, which may lead to underperformance of the teacher model. We evaluate two settings: (1) Homogeneous Structure, where the teacher and student share the same architecture type (e.g., ResNet-32x4 and ResNet-8x4), and (2) Heterogeneous Structure, where the teacher and student use different architectures (e.g., ResNet-32x4 and ShuffleNet-V1). For further details of the models evaluated and the dataset, please refer to Appendix A.2. To quantify the prediction uncertainty of the pretrained teacher model $f_p$, we utilize the RAPS algorithm (Angelopoulos et al., 2020). Given an input image $x$, we obtain the prediction set $\mathcal{C}(x)$ with $\alpha = 0.1$ and define the uncertainty as $u(x) = \frac{|\mathcal{C}(x)|-1}{K-1}$ (Vovk et al., 2016), where $K = 100$ is the total number of classes. The adaptive weight is computed as $w = \exp(-\gamma u)$ with $\gamma = 10.0$, as described in Section 3. Please see Appendix A.2.3 for an detailed analysis of the design choices for the hyperparameters. We follow the same experimental settings as in previous work (Tian et al., 2019; Sun et al., 2024) for the coefficients $\lambda_\text{task}$ and $\lambda_\text{guide}$, as well as other training details.

BASELINES. We measure Top-1 classification accuracy for a range of baselines, including classic distillation methods, KD (Hinton, 2015), FitNet (Romero et al., 2014), PKT (Passalis and Tefas, 2018), FT (Kim et al., 2018), and LS-KD (Sun et al., 2024), evaluating each both with and without our approach AdaConG, alongside uncertainty-aware adaptations such as EA-KD (Su et al., 2025) and PTLoss (Zhang et al., 2024), as well as KD leveraging heuristic confidence estimators including maximum softmax probability (MSP) (Pearce et al., 2021), Monte Carlo (MC) dropout (Gal and Ghahramani, 2016), and output entropy (Namdari and Li, 2019).

Table 1: Top-1 accuracy (%) of various knowledge distillation methods on CIFAR-100 under homogeneous structure where teacher models underperform due to domain shift. $\Delta$ indicates performance gain over the base method. Following the protocol in (Sun et al., 2024), we highlight in orange $\Delta$ greater than 0.15, indicating non-trivial enhancement. We observe up to **+10.89%** higher accuracy.

| Teacher model | ResNet110 | ResNet56 | ResNet32×4 | VGG13 | WRN-40-2 | WRN-40-2 |
|---|---|---|---|---|---|---|
| Teacher accuracy | 58.78 | 56.23 | 62.61 | 61.47 | 58.76 | 58.76 |
| Student model | ResNet20 | ResNet20 | ResNet8×4 | VGG8 | WRN-40-1 | WRN-16-2 |
| Student accuracy (from scratch) | 66.51±0.14 | 66.51±0.14 | 69.14±0.21 | 67.18±0.18 | 69.03±0.21 | 70.34±0.20 |
| KD (Hinton, 2015) | 57.23±0.24 | 56.27±0.17 | 58.90±0.31 | 61.00±0.25 | 58.44±0.16 | 59.40±0.33 |
| KD + AdaConG | 66.53±0.55 | 66.98±0.25 | 68.45±0.29 | 67.53±0.18 | 69.31±0.25 | 70.29±0.39 |
| $\Delta$ | 9.30 | 10.71 | 9.45 | 6.53 | 10.87 | 10.89 |
| FitNet (Romero et al., 2014) | 64.65±0.30 | 64.98±0.16 | 69.21±0.17 | 67.19±0.39 | 68.74±0.24 | 70.49±0.27 |
| FitNet + AdaConG | 67.06±0.13 | 66.91±0.14 | 69.49±0.18 | 67.58±0.30 | 69.11±0.18 | 71.00±0.23 |
| $\Delta$ | 2.41 | 1.93 | 0.28 | 0.39 | 0.37 | 0.51 |
| PKT (Passalis and Tefas, 2018) | 66.67±0.17 | 66.54±0.26 | 69.69±0.34 | 67.06±0.09 | 69.12±0.22 | 70.55±0.26 |
| PKT + AdaConG | **67.55**±0.11 | **67.42**±0.51 | 70.27±0.39 | 68.50±0.13 | 70.03±0.51 | 71.22±0.47 |
| $\Delta$ | 0.88 | 0.88 | 0.58 | 1.44 | 0.91 | 0.67 |
| FT (Kim et al., 2018) | 66.47±0.09 | 66.05±0.40 | 69.55±0.33 | 67.28±0.17 | 68.05±0.42 | 69.86±0.24 |
| FT + AdaConG | 66.58±0.12 | 66.55±0.39 | 69.85±0.23 | 67.54±0.19 | 69.03±0.33 | 70.91±0.30 |
| $\Delta$ | 0.11 | 0.50 | 0.30 | 0.26 | 0.98 | 1.05 |
| LS-KD (Sun et al., 2024) | 63.38±0.29 | 62.66±0.29 | 63.49±0.06 | 66.66±0.10 | 65.72±0.10 | 66.58±0.17 |
| LS-KD + AdaConG | 67.17±0.08 | 67.28±0.18 | **70.33**±0.14 | **68.99**±0.23 | **69.80**±0.18 | **71.48**±0.28 |
| $\Delta$ | 3.79 | 4.62 | 6.84 | 2.33 | 4.08 | 4.90 |
| Entropy (Namdari and Li, 2019) | 60.24±0.23 | 60.29±0.19 | 62.37±0.23 | 64.37±0.29 | 62.47±0.21 | 63.17±0.23 |
| MC dropout (Gal and Ghahramani, 2016) | 63.59±0.13 | 63.42±0.27 | 67.94±0.05 | 67.90±0.36 | 69.64±0.11 | 70.23±0.24 |
| EA-KD (Su et al., 2025) | 66.30±0.20 | 66.52±0.30 | 69.04±0.15 | 67.05±0.07 | 69.37±0.23 | 70.33±0.20 |
| MSP (Pearce et al., 2021) | 60.71±0.19 | 60.62±0.16 | 62.77±0.36 | 64.56±0.23 | 62.80±0.22 | 63.70±0.10 |
| PTLoss (Zhang et al., 2024) | 65.96±0.18 | 66.28±0.16 | 68.27±0.02 | 66.52±0.19 | 68.59±0.21 | 69.35±0.31 |

EXPERIMENTAL RESULTS. We present the results in Table 1. Following the protocol outlined in (Sun et al., 2024), we highlight in orange the improvements greater than 0.15, indicating non-trivial enhancements. Traditional knowledge distillation methods typically assume that the teacher model is reliable and superior to the student. However, as shown in Table 1, when the teacher performs worse than the student under domain shift, following the teacher will result in students that perform worse than those trained from scratch. In contrast, integrating AdaConG not only improves model

performance by up to 10.89% but also enables the student to surpass that from scratch. These findings highlight `AdaConG`'s effectiveness in enhancing supervised learning, leveraging a pretrained model as guidance, even when the guidance is unreliable. By selectively leveraging useful "dark knowledge" while avoiding misleading supervision, `AdaConG` ensures robust model learning.

We also compare against other uncertainty-aware KD methods (Su et al., 2025; Zhang et al., 2024; Pearce et al., 2021; Namdari and Li, 2019; Gal and Ghahramani, 2016). Results show that KD combined with `AdaConG` outperforms these baselines. This is because existing methods primarily rely on heuristic uncertainty estimates, which can be overconfident and poorly calibrated, particularly under domain shifts. In contrast, `AdaConG` can provide more adaptive and reliable guidance. Moreover, methods such as MC dropout require multiple forward passes, which is computationally expensive. Please refer to Appendix A.2.10 for a comparison of the computation overhead between `AdaConG` and MC dropout.

ABLATION STUDIES. We evaluate the performance of a heterogeneous teacher-student framework and present the results in Table 5 in Appendix A.2.2. The results show that, for all knowledge distillation methods, performance improves when combined with `AdaConG`, further validating the effectiveness of our approach across different teacher-student structures.

We also explore another hard version of the weighting function: $w = 1$ if $u = 0$, $w = 0$ if $u > 0$ (Vovk et al., 2016), which is similarly effective, please see the results in Appendix A.2.4. The rationale for the hard weighting function follows (Vovk et al., 2016), which aims to ensure that prediction sets are as close as possible to single-element sets, making them more informative.

**Semi-Supervised Image Classification.** We evaluate the effectiveness of our framework in improving the performance of semi-supervised learning methods, regarding classification tasks.

EXPERIMENTAL SETUP. We conduct experiments on several SSL image classification benchmarks, including CIFAR-10/100 (Krizhevsky et al., 2009) and STL-10 (Coates et al., 2011). For all experiments, we report the mean and standard deviation over four repeated runs. We measure the Top-1 accuracy for a range of baselines, including UDA (Xie et al., 2020), FixMatch (Sohn et al., 2020) and FlexMatch (Zhang et al., 2021), evaluating each both with and without `AdaConG`. For an unlabeled image $x$, we construct the prediction set $\mathcal{C}(x)$ for pseudo-labeling using confidence score as described in Section 3 with $\alpha = 0.05$. The associated uncertainty is defined as $u(x) = \frac{|\mathcal{C}(x)|-1}{K-1}$ (Vovk et al., 2016), where $K$ is the total number of classes. The adaptive weight is given by $w = \exp(-\gamma u)$ with $\gamma = 8.0$. Please refer to Appendix A.3.2 for a sensitivity analysis of the hyperparameter choices and Appendix A.3.1 for the training details.

Table 2: Top-1 accuracy (%) of various baselines with and without `AdaConG` on several semi-supervised image classification benchmarks, using cross-entropy as the guidance loss. $\Delta$ shows mean performance gain w.r.t. conventional methods without `AdaConG`, upto **+5.98%** in accuracy.

| Approach | CIFAR-10 | | | CIFAR-100 | | | STL-10 | | |
|---|---|---|---|---|---|---|---|---|---|
| | 40 labels | 250 labels | 4000 labels | 400 labels | 2500 labels | 10000 labels | 40 labels | 250 labels | 1000 labels |
| UDA (Xie et al., 2020) | 57.73±6.98 | 89.44±1.57 | 91.86±0.75 | 25.95±1.07 | 57.57±0.12 | 66.94±0.16 | 53.88±0.58 | 76.91±0.10 | 87.59±0.34 |
| UDA + AdaConG | 61.28±6.33 | 92.69±0.12 | 93.17±0.10 | 28.02±0.54 | 58.54±0.80 | 67.50±0.35 | 54.70±0.51 | 77.45±0.12 | 88.31±0.10 |
| $\Delta$ | 3.55 | 3.25 | 1.31 | 3.07 | 0.97 | 0.56 | 0.82 | 0.54 | 0.72 |
| FixMatch (Sohn et al., 2020) | 64.18±4.57 | 89.97±1.04 | 91.29±0.65 | 40.36±0.83 | 61.14±0.40 | 67.50±0.85 | 58.03±1.28 | 78.89±0.46 | 88.54±0.10 |
| FixMatch + AdaConG | 70.16±3.34 | 92.23±0.72 | 93.97±0.11 | 41.98±0.55 | 63.41±1.46 | 70.03±0.29 | 62.70±0.84 | 80.83±0.39 | 89.35±0.10 |
| $\Delta$ | 5.98 | 2.26 | 2.66 | 1.62 | 2.27 | 2.43 | 4.67 | 1.94 | 0.81 |
| FlexMatch (Zhang et al., 2021) | 73.24±1.61 | 90.62±0.49 | 92.11±0.47 | 51.25±1.63 | 63.59±1.03 | 71.63±0.48 | 62.55±2.22 | 82.63±1.20 | 89.94±0.20 |
| FlexMatch + AdaConG | 76.98±0.45 | 92.89±0.10 | 94.28±0.10 | 55.63±1.20 | 69.22±0.52 | 72.71±0.28 | 65.98±1.55 | 83.94±0.23 | 92.27±0.10 |
| $\Delta$ | 3.74 | 2.27 | 2.17 | 4.38 | 5.63 | 1.08 | 3.43 | 1.31 | 2.33 |

EXPERIMENTAL RESULTS. We present the results in Table 2. As shown, integrating `AdaConG` consistently improves performance across all baselines. This highlights the effectiveness of `AdaConG` in semi-supervised learning. By adaptively reweighting the influence of pseudo-labels, `AdaConG` reduces reliance on noisy supervision, mitigating error propagation and leading to improved overall performance.

As ablation studies, in addition to the commonly used cross-entropy loss for guidance in SSL, we investigate the mean-squared error (MSE) loss as an alternative. Specifically, we apply the MSE loss between the logits of a strongly augmented input $x_{\text{strong}}$ and its weakly augmented counterpart $x_{\text{weak}}$. As shown in Table 3, our method remains effective under this alternative formulation and continues to yield performance improvements.

**Gridworld Navigation.** We investigate the use of reinforcement learning for solving gridworld navigation tasks, leveraging a pretrained imitation policy as prior guidance. We demonstrate how `AdaConG` improves policy learning efficiency and robustness in challenging and unseen environments when the imitation policy is limited due to generalization constraints.

EXPERIMENTAL SETUP. We evaluate `AdaConG` across three gridworld environment (Chevalier-Boisvert et al., 2024) scenarios, as illustrated in Fig. 9. For the environment details, please refer to Appendix A.4. We collect expert demonstration for the Lava 1

Table 3: Top-1 accuracy (%) on CIFAR-100 for semi-supervised image classification. We compare multiple baselines with and without `AdaConG` using MSE as the guidance loss. $\Delta$ indicates the average improvement over each corresponding baseline.

| Approach | 400 labels | 2500 labels | 10000 labels |
|---|---|---|---|
| UDA (Xie et al., 2020) | $6.36 \pm 0.47$ | $31.48 \pm 0.38$ | $57.50 \pm 0.31$ |
| UDA+`AdaConG` | $7.76 \pm 0.15$ | $32.60 \pm 0.21$ | $59.79 \pm 0.11$ |
| $\Delta$ | 1.40 | 1.12 | 2.29 |
| FixMatch (Sohn et al., 2020) | $8.56 \pm 0.26$ | $33.14 \pm 0.69$ | $60.93 \pm 0.41$ |
| FixMatch+`AdaConG` | $9.82 \pm 0.62$ | $35.36 \pm 1.03$ | $61.74 \pm 0.20$ |
| $\Delta$ | 1.26 | 2.22 | 0.81 |
| FlexMatch (Zhang et al., 2021) | $10.04 \pm 0.24$ | $36.10 \pm 0.36$ | $61.36 \pm 0.07$ |
| FlexMatch+`AdaConG` | $11.07 \pm 0.55$ | $38.87 \pm 0.82$ | $62.80 \pm 0.21$ |
| $\Delta$ | 1.03 | 2.77 | 1.44 |

and Door environments to train IL policies via behavior cloning. After training the IL policy $\pi_I$, we utilize it to guide the training of the RL policy $\pi_R$. The Lava 2 environment represents a shifted variant of Lava 1, featuring modified environmental configurations. Importantly, we do not collect expert demonstration for Lava 2, and no IL policy is trained on this environment. For a given state $s$, the guidance weight is defined as: $w(s) = \frac{\exp(-u_I(s))}{\exp(-u_I(s))+\exp(-u_R(s))}$, where $u_I$ and $u_R$ are the prediction uncertainties of the IL and RL policies as described in Section 3. And we sample the action $a \in \{a_I, a_R\}$ to take according to the distribution induced by this guidance weight. Furthermore, we explore another hard variant of `AdaConG`, instead of defining $w(s)$ as a probability distribution informed by the relative uncertainties of the IL and RL policies, we take argmax to compare IL and RL prediction uncertainties: $w(s) = 1$ when $u_I(s) < u_R(s)$, otherwise $w(s) = 0$. Based on $w(s)$, we dynamically decide which action to take: $a = a_I$ if $w(s) = 1$, otherwise $a = a_R$.

BASELINES. We compare `AdaConG` and Hard `AdaConG` against several baselines, including (1) **Soft Actor Critic (SAC)** (Haarnoja et al., 2018), a purely RL approach, (2) **IBRL** (Hu et al., 2023), which leverages a pretrained imitation learning (IL) model to bootstrap RL. During the training process, IBRL queries the target Q-network and selects actions by comparing the Q-values of two candidate actions and taking the action with the higher Q-value, and (3) **Soft IBRL** (Hu et al., 2023), a probabilistic variant of IBRL. Instead of selecting the action via a hard argmax, Soft IBRL samples the action according to a distribution proportional to the Q-values.

EXPERIMENTAL RESULTS. We run all experiments across ten random seeds and present the results in Fig. 2. First, we compare the learning curves of `AdaConG` and Hard `AdaConG` against other baselines across three environments: Lava 1, Lava 2, and Door. Both `AdaConG` and Hard `AdaConG` demonstrate similar performance, converging faster and achieving higher rewards than all other baselines. Before the agent reaches the goal, the reward function is defined as the negative Manhattan distance between the agent's current location and the goal, normalized by the maximum step limit of 100. Consequently, the accumulated episode rewards initially decrease as the agent explores the environment and accrues negative rewards but increase as it learns. The rewards of `AdaConG` and Hard `AdaConG` are consistently higher than those of other baselines while converging faster. This can be attributed to the efficiency of `AdaConG`, as it compares the prediction uncertainties of teacher and student models instead of relying on Q-values. Methods like IBRL and Soft IBRL, which depend on Q-values to decide between IL or RL actions, may make suboptimal decisions initially due to poorly trained Q-networks. For instance, even if an IL action is superior, its Q-value might be lower than that of an RL action.

In the shifted environment Lava 2, the overall rewards of the IL policy are lower due to generalization constraints. IBRL and Soft IBRL rewards eventually converge close to the IL policy's performance, as these methods are not uncertainty-aware. Blindly relying on a IL policy underperforming due to environment shifts can lead to suboptimal performance. In contrast, `AdaConG` and Hard `AdaConG` consider the IL policy's prediction uncertainty, enabling faster convergence and achieving rewards over $6\times$ higher than the best-performing baselines after convergence. When the IL policy's predictions are confident, the RL agent relies more on them; otherwise, the RL agent explores independently. Even though the IL policy's overall reward is not high, it still provides use-

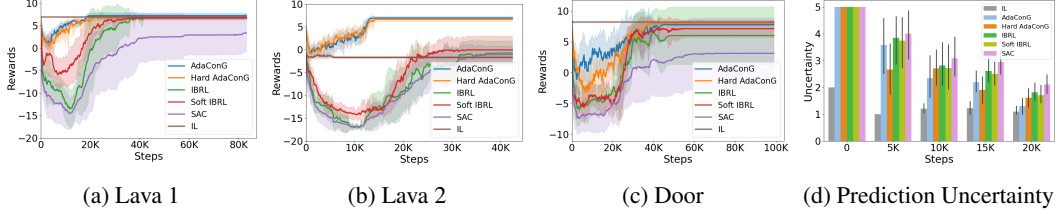

| (a) Lava 1 | (b) Lava 2 | (c) Door | (d) Prediction Uncertainty |

Figure 2: **(a-c) Learning Curves.** We compare `AdaConG` and Hard `AdaConG` with other baselines, including SAC, IBRL, and Soft IBRL, and present their learning curves across three environments: (a) Lava 1, (b) Lava 2, and (c) Door. `AdaConG` and Hard `AdaConG` perform similarly, converging faster and achieving higher rewards than other baselines in all environments. **(d) Prediction Uncertainty.** We show the average prediction uncertainties of `AdaConG` and other baselines, taking the Lava 1 environment as the example. Over time, the uncertainty of `AdaConG` shrinks and approach that of the IL policy, demonstrating the development of a well-learned RL policy. In addition, `AdaConG` maintains lower prediction uncertainty than other baselines, indicating stronger robustness to uncertainty.

ful knowledge. This allows the RL agent to learn from the IL policy and eventually achieve higher rewards than the IL policy.

We also show the average policy prediction uncertainties of `AdaConG` and Hard `AdaConG`, for Lava 1 environment, in Fig. 2d. Over time, their prediction uncertainties decrease and approach that of the IL policy, demonstrating the progression toward a well-learned RL policy.

**Autonomous Driving.** This task involves learning a steering prediction policy in autonomous driving for an RGB-only input model, guided by a pretrained multi-modal teacher to transfer knowledge to the student model. We evaluate the effectiveness of different knowledge distillation methods under domain shifts and sensor noise, comparing performance with and without the use of `AdaConG`.

EXPERIMENTAL SETUP. We adopt mean accuracy (mAcc) as the evaluation metric for the task of steer prediction, following prior works (Shen et al., 2021; 2023; 2024). We use the real-world driving dataset SullyChen (Chen, 2018) for evaluation, which includes diverse driving scenarios with various road types and conditions. We use Nvidia PilotNet (Bojarski, 2016) as the backbone for both the teacher and student models. The teacher model is a multi-

Table 4: Mean accuracy (%) of steer prediction of different knowledge transfer methods with and without `AdaConG` under domain shifts.

| Approach | Mean Accuracy (%) | | |
|---|---|---|---|
| | without `AdaConG` | with `AdaConG` | Δ |
| KD | 73.5 | 76.8 | 3.3 |
| FitNet | 72.4 | 76.2 | 3.8 |
| PKT | 72.8 | 75.9 | 3.1 |
| FT | 73.1 | 76.4 | 3.3 |
| Teacher (RGB+Depth+Edge) Acc. | 78.5 | – | – |
| Student (RGB) Acc. from scratch | 71.8 | – | – |

modal network that takes RGB images, depth, and edge maps as input, while the student model is unimodal, relying solely on RGB images. For more details of the setup, please see Appendix A.5.1. We first train the teacher model $f_p$ offline. Then we use it to guide the student model $f_t$ learning, while the RGB images for $f_t$ training have domain shifts by Gaussian noise corruption compared to the ones used for $f_p$ training. Detailed information about domain shifts and model training can be found in Appendices A.5.2 and A.5.4, respectively. We evaluate multiple knowledge distillation methods as baselines, comparing their performance with and without the integration of `AdaConG`, including KD (Hinton, 2015), FitNet (Romero et al., 2014), PKT (Passalis and Tefas, 2018), and FT (Kim et al., 2018).

EXPERIMENTAL RESULTS. We report the mean accuracy of steer prediction for various KD methods under domain shifts with and without `AdaConG` in Table 4. As observed, incorporating `AdaConG` consistently enhances the accuracy. This demonstrates the effectiveness of `AdaConG` in improving model performance, as its adaptive guidance mechanism strategically prevents over-reliance on uncertain teacher predictions, facilitating more reliable and robust target model learning.

## 5 CONCLUSIONS

We propose `AdaConG`, an approach for learning with guidance under uncertainty. `AdaConG` integrates split conformal prediction to adaptively modulate the influence of guidance signals based on their associated uncertainty. By selectively leveraging reliable signals and filtering out misleading supervision, `AdaConG` enables effective learning even in the presence of noise. Unlike conven-

tional methods that assume guidance is always trustworthy, `AdaConG` can still extract useful "dark knowledge" under uncertainty. The framework is simple yet effective, and broadly applicable to a wide range of tasks. We validate `AdaConG` across diverse settings and tasks including knowledge distillation, semi-supervised image classification, gridworld navigation, and autonomous driving, demonstrating improved performance and robustness. For a discussion on future work, please see Appendix A.6.

## 6 REPRODUCIBILITY STATEMENT

To ensure reproducibility, we provide detailed descriptions of all training setups in the experiment Section 4, with additional specifications in Appendix A.2, Appendix A.3.1, Appendix A.4, and Appendix A.5.1. All datasets used are publicly available.

## ACKNOWLEDGEMENTS

This work is supported in part by Dr. Barry Mersky and Capital One E-Nnovte Endowed Professorships, University of Maryland Distinguished University Professorship, and ARL-UMD ArtIAMAS Cooperative Agreement.

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

## A APPENDIX

### A.1 THE USE OF LLMS

We use LLMs to polish writing and refine grammar of the manuscript.

### A.2 KNOWLEDGE DISTILLATION

We evaluate two settings: (1) Homogeneous Structure, where both the teacher and student models share the same type of architecture (e.g., ResNet-32x4 and ResNet-8x4), and (2) Heterogeneous Structure, where the teacher and student models are of different architectures (e.g., ResNet-32x4 and ShuffleNet-V1). We evaluate a wide range of neural network architectures, including ResNet (He et al., 2016), WRN (Zagoruyko, 2016), VGG (Simonyan, 2014), ShuffleNet-V1 (Zhang et al., 2018)/V2 (Ma et al., 2018), and MobileNet-V2 (Sandler et al., 2018).

#### A.2.1 DATASET DETAILS

We conduct our experiments on the CIFAR-100 dataset Krizhevsky et al. (2009), which consists of 60K images, 50K for training and 10K for testing, across 100 distinct categories. We introduce domain shifts to the dataset for training the target model. We add Gaussian noise with zero mean and a standard deviation of 0.05 to $40\%$ of the training data of 50K images, where the noisy samples are selected uniformly at random across the entire dataset to ensure consistent noise distribution. Then we shuffle the dataset and randomly split it into a $90\%$ training set $\mathcal{D}_{\text{train}}^t$, and a $10\%$ calibration set $D_{\text{cal}}$, ensuring that both sets are drawn from the same underlying distribution. Additionally, the same Gaussian noise is added to $40\%$ of the testing data of 10K images, to form the noisy test set $D_{\text{test}}$, which allows us to evaluate the performance of the target model.

#### A.2.2 HETEROGENEOUS TEACHER-STUDENT STRUCTURE

Table 5: Top-1 accuracy (%) of various knowledge distillation methods with and without `AdaConG` on CIFAR-100 under heterogeneous structure. We use $\Delta$ to show mean performance gain relative to conventional knowledge distillation methods without `AdaConG`. We highlight in orange deltas greater than 0.15, indicating non-trivial enhancement following the protocol in Sun et al. (2024).

| Teacher | ResNet50 | VGG13 | WRN-40-2 |
|---|---|---|---|
| | 62.79 | 61.47 | 58.76 |
| Student | ShuffleNet-V1 | MobileNet-V2 | ShuffleNet-V2 |
| | 64.52±0.62 | 56.47±0.03 | 66.35±0.12 |
| KD Hinton (2015) | 58.30±0.24 | 53.08±0.57 | 59.61±0.03 |
| KD + `AdaConG` | 65.71±0.36 | 57.69±0.52 | 67.57±0.22 |
| $\Delta$ | 7.41 | 4.41 | 7.96 |
| FitNet Romero et al. (2014) | 63.97±0.25 | 54.77±0.40 | 66.03±0.48 |
| FitNet + `AdaConG` | 64.49±0.15 | 55.75±0.37 | 67.69±0.11 |
| $\Delta$ | 0.52 | 0.98 | 1.66 |
| PKT Passalis and Tefas (2018) | 66.26±0.26 | 56.53±0.13 | 66.38±0.18 |
| PKT + `AdaConG` | 66.65±0.16 | 57.01±0.18 | 67.88±0.42 |
| $\Delta$ | 0.39 | 0.48 | 1.50 |
| FT Kim et al. (2018) | 63.85±0.34 | 56.36±0.49 | 66.34±0.21 |
| FT + `AdaConG` | 65.13±0.43 | 57.40±0.20 | 67.29±0.12 |
| $\Delta$ | 1.28 | 1.04 | 0.95 |

As part of our ablation studies, we evaluate the performance of a heterogeneous teacher-student framework and present the results in the following Table 5. The table shows that, for all knowledge transfer methods, performance improves when combined with `AdaConG`, further validating the effectiveness of our approach across different teacher-student structures.

#### A.2.3 HYPERPARAMETER SENSITIVITY ANALYSIS

We conduct sensitivity analysis for the design choices of the key hyperparameters, including the error rate $\alpha$ for split CP and the temperature $\gamma$ of the adaptive weighting $w = \exp(-\gamma u)$. We use ResNet-110 as the pretrained teacher and ResNet-20 as the student model to train.

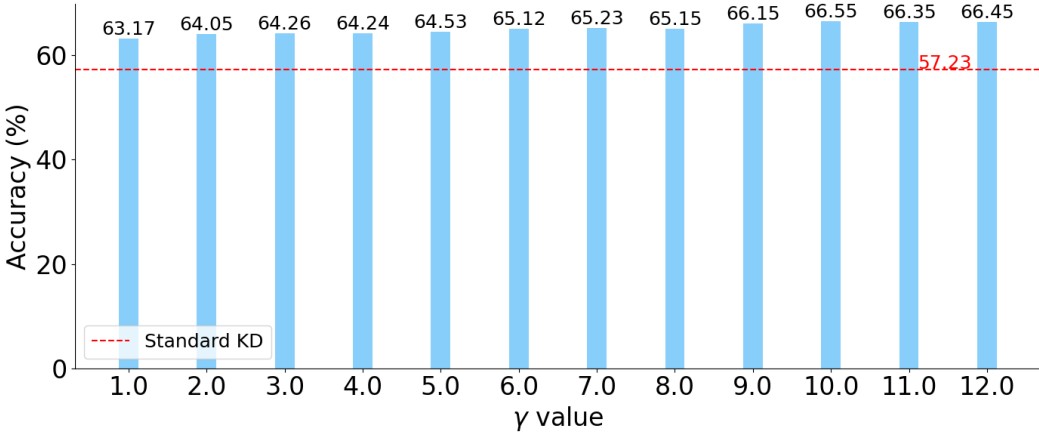

Figure 3: Top-1 accuracy of KD using `AdaConG` with varying temperature $\gamma$ values for adaptive weighting.

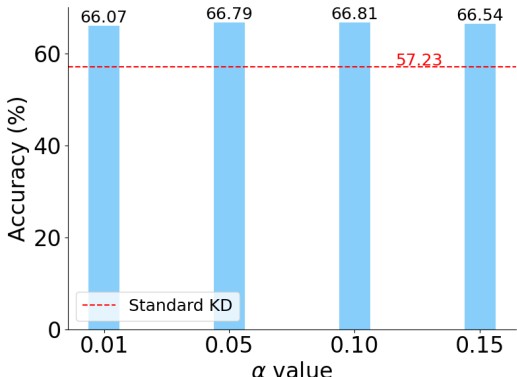

Figure 4: Top-1 accuracy of KD using `AdaConG` with varying $\alpha$ values. The results demonstrate that our approach is robust to the choice of $\alpha$ and consistently outperforms standard KD.

We first analyze the sensitivity of the temperature parameter $\gamma$ in the adaptive weighting function. The results are presented in Fig. 3, showing how $\gamma$ affects the student's Top-1 accuracy with $\alpha = 0.1$. Across all tested values, combining KD with `AdaConG` consistently outperforms standard KD. We observe that accuracy generally increases as $\gamma$ increases. This trend is intuitive, when $\gamma$ is too small, the exponential decay used for reweighting may not sufficiently suppress the influence of noisy teacher predictions. We select $\gamma = 10.0$ as our default setting, since performance tends to not increase for $\gamma > 10.0$.

We then analyze the sensitivity of $\alpha$. We present the results in Fig. 4, showing how $\alpha$ influences the student's Top-1 accuracy with $\gamma = 10.0$. The results demonstrate that our approach is robust to the choice of $\alpha$ and consistently outperforms standard KD. We choose $\alpha = 0.1$ which yields slightly better results. The underlying insight is as follows: when $\alpha$ is small, the prediction set becomes large, indicating lower teacher confidence. As a result, the teacher's guidance becomes less informative, and the student relies more on its own learning, reducing the benefit of distillation, even when the teacher is accurate. Conversely, when $\alpha$ is large, the prediction set shrinks, making the teacher appear overly confident. In this case, the student may rely too heavily on potentially noisy teacher predictions, leading to suboptimal knowledge transfer. We further support this interpretation by presenting the average size of the prediction set for the teacher model in Fig. 5, which aligns with our observations.

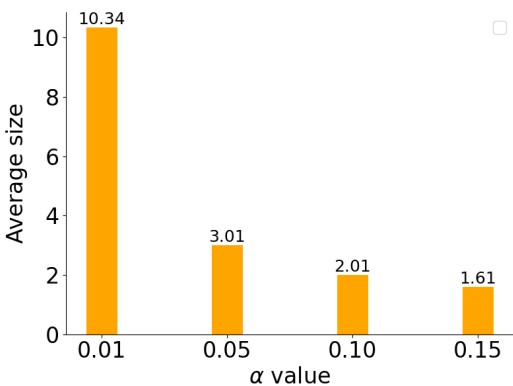

Figure 5: Average size of the prediction set for the teacher model.

### A.2.4  HARD WEIGHTING FUNCTION

We explore another hard version of weighting function: $w = 1$ if $u = 0$, $w = 0$ if $u > 0$ (Vovk et al., 2016). We use ResNet-110 as the pretrained teacher and ResNet-20 as the student model to train. The rationale for the hard weighting function follows (Vovk et al., 2016), which aims to ensure that prediction sets are as close as possible to single-element sets, making them more informative. This scheme is simple and effective, allowing the student to learn from high-confidence teacher predictions while filtering out potentially misleading guidance. Experimental results in Table 6 show that combining this hard weighting scheme with `AdaConG` outperforms standard knowledge distillation methods.

Table 6: Top-1 accuracy (%) of various knowledge distillation methods without and with `AdaConG` using the hard weighting function. We use $\Delta$ to show performance gain relative to conventional knowledge distillation methods and highlight in orange deltas greater than 0.15, indicating non-trivial enhancement following the protocol in (Sun et al., 2024).

| Approach | Accuracy (%) |
|---|---|
| KD (Hinton, 2015) | 57.21 |
| KD + `AdaConG` | 66.52 |
| $\Delta$ | 9.31 |
| FitNet (Romero et al., 2014) | 64.65 |
| FitNet + `AdaConG` | 66.88 |
| $\Delta$ | 2.23 |
| PKT (Passalis and Tefas, 2018) | 66.50 |
| PKT + `AdaConG` | 66.83 |
| $\Delta$ | 0.33 |
| FT (Kim et al., 2018) | 66.47 |
| FT + `AdaConG` | 66.69 |
| $\Delta$ | 0.22 |
| LS-KD (Sun et al., 2024) | 63.40 |
| LS-KD + `AdaConG` | 67.11 |
| $\Delta$ | 3.71 |

### A.2.5  DIRECT USE OF NONCONFORMITY SCORES

We conduct additional experiments to compare the performance of directly using nonconformity scores versus applying `AdaConG` with quantile computation, as ablation studies, across different teacher/student setups with KD (Hinton, 2015). The results are shown in Table 7, reporting the mean and standard deviation over four repeated runs. As shown, `AdaConG` outperforms the direct use of nonconformity scores. This indicates that computing the quantile of the nonconformity scores is helpful, it is necessary to construct the prediction set for split CP, therefore transferring non conformalized uncertainty measures into rigorous ones.

Table 7: Top-1 accuracy (%) on CIFAR-100 for the KD approach, comparing direct use of nonconformity scores versus AdaConG using quantile computation. `AdaConG` outperforms the direct use of nonconformity scores.

| Approach | ResNet110/ResNet20 | ResNet56/ResNet20 | ResNet32x4/ResNet8x4 | VGG13/VGG8 | WRN-40-2/WRN-40-1 |
|---|---|---|---|---|---|
| Nonconformity score | $64.17 \pm 0.41$ | $64.90 \pm 0.28$ | $66.89 \pm 0.15$ | $65.19 \pm 0.34$ | $68.06 \pm 0.13$ |
| AdaConG | $66.53 \pm 0.55$ | $66.98 \pm 0.25$ | $68.45 \pm 0.29$ | $67.53 \pm 0.18$ | $69.31 \pm 0.25$ |

### A.2.6 EXPERIMENTS ON CIFAR-100-C DATASET

We conduct additional experiments on the more challenging CIFAR-100-C benchmark (Hendrycks and Dietterich, 2019). We evaluate under the Gaussian noise corruption with five levels of severity, larger severity corresponds to stronger noise and greater distribution shift. More details can be found in (Hendrycks and Dietterich, 2019). We use two pretrained teacher models, ResNet-110 and ResNet-56, both trained on the clean CIFAR-100 dataset, and perform knowledge distillation under different methods, including KD (Hinton, 2015) and LS-KD (Sun et al., 2024), with both homogeneous and heterogeneous teacher–student structures.

The results in Table 8 show that incorporating `AdaConG` consistently improves student performance across all settings. These gains further demonstrate that our method effectively modulates imperfect teacher signals under distribution shift and enhances learning performance.

Table 8: Top-1 accuracy (%) of different knowledge distillation methods on CIFAR-100-C. $\Delta$ indicates performance gain over the base method. We observe up to +18.62% higher accuracy improvement.

| Teacher model | ResNet110 | ResNet56 | ResNet110 | ResNet56 |
|---|---|---|---|---|
| Teacher accuracy | 24.84 | 20.61 | 24.84 | 20.61 |
| Student model | ResNet20 | ResNet20 | ShuffleNet-V1 | ShuffleNet-V2 |
| Student accuracy (from scratch) | $41.24 \pm 0.55$ | $41.24 \pm 0.55$ | $35.43 \pm 0.68$ | $35.56 \pm 0.61$ |
| KD (Hinton, 2015) | $24.32 \pm 0.36$ | $20.54 \pm 0.53$ | $24.60 \pm 0.17$ | $20.58 \pm 0.27$ |
| KD + AdaConG | $42.16 \pm 0.36$ | $39.16 \pm 0.43$ | $35.57 \pm 0.73$ | $37.22 \pm 0.49$ |
| $\Delta$ | 17.84 | 18.62 | 10.97 | 16.64 |
| LS-KD (Sun et al., 2024) | $41.02 \pm 0.52$ | $38.58 \pm 0.08$ | $33.93 \pm 0.70$ | $34.94 \pm 0.92$ |
| LS-KD + AdaConG | $44.44 \pm 0.38$ | $43.33 \pm 0.21$ | $37.76 \pm 0.33$ | $40.34 \pm 0.63$ |
| $\Delta$ | 3.42 | 4.75 | 3.83 | 5.40 |

### A.2.7 EXPERIMENTS ON LARGER DATASET AND LARGER BACKBONE

We also conduct additional experiments on larger dataset and larger backbone to further validate our approach. Specifically, we use a larger dataset Tiny ImageNet (Le and Yang, 2015), which contains 100K training images and 10K testing images across 200 classes. For the teacher model, we adopt a larger backbone ViT-B (Dosovitskiy et al., 2020) and train it on the clean Tiny ImageNet dataset for 50 epochs. During knowledge distillation to different student models, we introduce noise into the Tiny ImageNet dataset following the procedure described in Appendix A.2.1. This places the pretrained teacher model under distribution shift, making its prediction noisy and unreliable. In this setting, we leverage `AdaConG` to dynamically modulate the guidance from the imperfect teacher. The results in Table 9 show that incorporating `AdaConG` consistently improves student performance across both backbone scales, demonstrating that our approach remains effective with larger dataset and larger backbone.

### A.2.8 QUALITATIVE ANALYSIS

In this section, we present a qualitative analysis demonstrating how `AdaConG` assigns higher or lower uncertainty to different inputs, providing deeper insight into its behavior. We sample multiple clean images from the CIFAR-100 dataset and generate corresponding noisy images by adding Gaussian noise with zero mean and standard deviation 0.05. For both clean and noisy images, we apply a pretrained ResNet-110 model with conformal prediction to obtain the prediction set for each image. We then convert the size of each prediction set into an uncertainty score using Eq. 1, then an

Table 9: Top-1 accuracy (%) of different knowledge distillation methods on Tiny ImageNet. $\Delta$ indicates performance gain over the base method. We observe up to +19.02% higher accuracy.

| Teacher model | ViT-B | ViT-B | ViT-B |
|---|---|---|---|
| Teacher accuracy | 33.06 | 33.06 | 33.06 |
| Student model | ResNet20 | ShuffleNet-V1 | ShuffleNet-V2 |
| Student accuracy (from scratch) | 43.50±0.15 | 50.08±0.21 | 51.79±0.11 |
| KD (Hinton, 2015) | 31.21±0.14 | 32.87±0.33 | 33.00±0.24 |
| KD + `AdaConG` | 43.42±0.32 | 51.20±0.21 | 52.02±0.28 |
| $\Delta$ | 12.21 | 18.33 | 19.02 |
| LS-KD (Sun et al., 2024) | 42.16±0.39 | 48.59±0.16 | 49.76±0.17 |
| LS-KD + `AdaConG` | 45.50±0.28 | 52.65±0.41 | 54.69±0.28 |
| $\Delta$ | 3.34 | 4.06 | 4.93 |

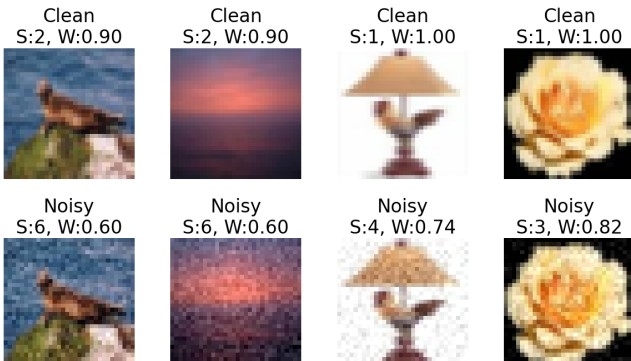

Figure 6: **Qualitative analysis of adaptive weights**. The first row shows original clean images from CIFAR-100, and the second row shows their noisy counterparts. Each image is annotated with the prediction set size (S) and the corresponding adaptive weight (W). Clean images produce smaller prediction sets, indicating lower uncertainty and thus higher weights, while noisy images yield larger prediction sets, reflecting higher uncertainty and consequently lower weights.

adaptive weight using Eq. 2. We present the results in Figure 6, the first row contains the original clean images, while the second row shows their noisy counterparts. For each image, we annotate the prediction set size (S) and the obtained adaptive weight (W). As shown, clean images yield smaller prediction sets and thus higher adaptive weights, whereas noisy images produce larger prediction sets, indicating higher uncertainty and consequently lower adaptive weights for the guidance signal.

### A.2.9 TRAINING DETAILS

For the experiments, we use the stochastic gradient descents (SGD) (Sutskever et al., 2013) as the optimizer with momentum 0.9 and weight decay $5e-4$. The epoch number is 240 and the batch size is 128. The initial learning rate is set to 0.01 for MobileNet (Sandler et al., 2018)/ShuffleNet (Zhang et al., 2018) architectures and 0.05 for other architectures. The model is trained on an Nvidia RTX 3090 GPU with AMD Ryzen 9 5900 CPU and 32 GB RAM.

### A.2.10 COMPUTATION OVERHEAD

We compare the training cost of our approach `AdaConG` against standard knowledge distillation (KD) and KD with MC dropout on an RTX 3090 GPU. For MC dropout, we perform ten forward passes and average the outputs. Compared to MC dropout, the training cost of using `AdaConG` is much lower. Compared to standard KD, the training cost of `AdaConG` is close, since using split CP is just a single pass, as shown in Table 10. From the results in Table 10, using `AdaConG` just adds around 0.003 ms overhead per sample during training.

Table 10: Comparison of computational overhead between standard KD, KD with `AdaConG`, and KD with MC dropout.

| Approach | Time/Epoch (s) |
|---|---|
| KD | 6.87 |
| KD + `AdaConG` | 7.04 |
| KD + MC dropout | 44.90 |

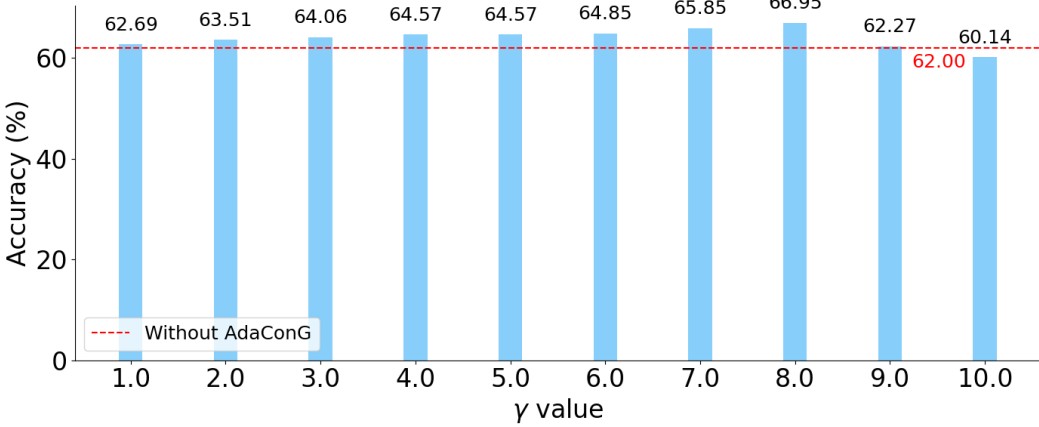

Figure 7: Prediction accuracy of FixMatch using `AdaConG` with varying temperature $\gamma$ values for adaptive weighting.

### A.3 SEMI-SUPERVISED IMAGE CLASSIFICATION

#### A.3.1 TRAINING DETAILS

Following the setup in Sohn et al. (2020); Zhang et al. (2021), we use the WRN-28-8 architecture Zagoruyko (2016) for the CIFAR-10 and CIFAR-100 datasets Krizhevsky et al. (2009), and WRN-37-2 Zagoruyko (2016) for the STL-10 dataset Coates et al. (2011). We adopt stochastic gradient descent (SGD) Sutskever et al. (2013) as the optimizer with a momentum of 0.9. The weight decay is set to $5 \times 10^{-4}$ for CIFAR-10 and STL-10, and $1 \times 10^{-3}$ for CIFAR-100. Models are trained for 51,200 iterations with a batch size of 64 and an initial learning rate of 0.03. Experiments are conducted on an Nvidia RTX 3090 GPU with AMD Ryzen 9 5900 CPU and 32 GB RAM.

#### A.3.2 HYPERPARAMETER SENSITIVITY ANALYSIS

We conduct sensitivity analysis for the design choices of the key hyperparameters, including $\alpha$ for split CP and the temperature $\gamma$ for the adaptive weighting $w = \exp(-\gamma u)$. We conduct experiments using the approach FixMatch (Sohn et al., 2020) combined with `AdaConG` on the CIFAR-10 dataset with 40 labels.

We first analyze the sensitivity of the temperature parameter $\gamma$ in the adaptive weighting function. The results are presented in Fig. 3, showing how $\gamma$ affects the prediction accuracy with $\alpha = 0.1$. Across all tested values, accuracy generally increases as $\gamma$ increases, which is expected, when $\gamma$ is too small, the exponential decay used for reweighting may not sufficiently suppress the influence of noisy pseudo-labels. We select $\gamma = 8.0$, as performance begins to slightly decline when $\gamma > 8.0$.

We then analyze the sensitivity of $\alpha$. The results, shown in Fig.8, illustrate how $\alpha$ affects prediction accuracy with $\gamma = 8.0$. FixMatch combined with `AdaConG` outperforms the standard approach across all values. The accuracy first increases and then decreases as $\alpha$ increases, with the highest performance achieved at $\alpha = 0.05$. Therefore, we select $\alpha = 0.05$. The underlying insights are similar to those observed in the knowledge distillation experiments.

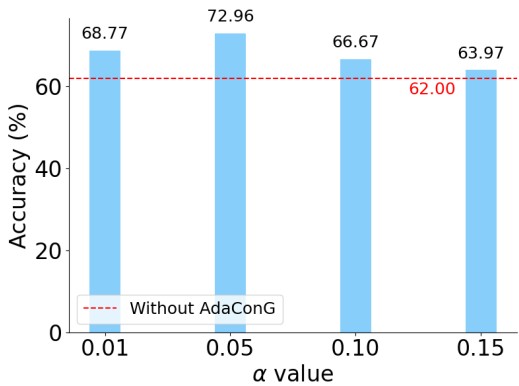

Figure 8: Prediction accuracy of FixMatch using `AdaConG` with varying $\alpha$ values.

### A.3.3 COMPUTATION OVERHEAD

We compare the training cost of the key SSL baseline FlexMatch with and without our method `AdaConG` on the CIFAR-100 dataset. As shown in Table 11, incorporating `AdaConG` introduces only minimal additional computational overhead.

Table 11: Comparison of computational overhead between FlexMatch and FlexMatch + `AdaConG`. Incorporating `AdaConG` adds only minimal computation.

| Approach | Time/Epoch (s) |
|---|---|
| FlexMatch | 91.52 |
| FlexMatch + `AdaConG` | 97.63 |

### A.4 IMITATION-GUIDED REINFORCEMENT LEARNING

We explore reinforcement learning for grid-world navigation, leveraging a pretrained imitation policy to provide prior guidance. The environment scenarios shown in Fig. 9 are adapted from (Yu et al., 2024) and developed using the Minigrid framework (Chevalier-Boisvert et al., 2024). They are fully observable with discrete state and action spaces. In each environment, the agent's state corresponds to its {x, y} coordinates on the map, and the action space comprises five discrete actions: left, right, up, down, and stay. Each episode is capped at a maximum of 100 steps. In the Lava 1 and Lava 2 environments, the reward function is computed as the negative Manhattan distance between the agent's current position and the goal, normalized by the maximum step limit of 100. Upon reaching the goal, the agent receives a terminal reward of $10 - 9 \times \frac{\text{step count}}{\text{max step}}$. Stepping into the lava results in a reward of -1, and the episode terminates immediately. Similarly, in the Door environment, the reward structure follows the same formulation but without lava, encouraging the agent to minimize its distance to the goal, with the same terminal reward applied upon successful completion.

We collect expert demonstration data comprising state-action pairs for the Lava 1 and Door environments to train imitation learning (IL) models via behavior cloning (Torabi et al., 2018). The demonstration data are inherently uncertain, as multiple valid actions may exist for the same state, as illustrated in (Yu et al., 2024), introducing ambiguity in the IL model's predictions.

We calibrate both the IL policy $\pi_I$ and the RL policy $\pi_R$, and estimate their prediction uncertainties $u_I$ and $u_R$ with $g$ as the identity mapping, $\alpha = 0.1$, $N = 1000$ and $m = 128$, as described in Section 3. For a given state $s$, the guidance weight is defined as: $w(s) = \frac{\exp(-u_I(s))}{\exp(-u_I(s)) + \exp(-u_R(s))}$. And we sample the action $a \in \{a_I, a_R\}$ to take according to the distribution induced by this guidance weight. To encourage exploration by the RL policy, we define a probability $\epsilon = \min(0.5\frac{t}{S_{\text{total}}} + 0.5\frac{e}{E_{\text{total}}}, 1)$, where $t$ is the current training step, $e$ is the current episode, $S_{\text{total}}$ and $E_{\text{total}}$ are the total training steps and episodes, respectively. At each step, if a random probability $p < \epsilon$, the agent takes an

action from the RL policy, otherwise it takes an action based on `AdaConG`. As $\epsilon$ increases over time, the agent progressively shifts to a learned RL policy, while initially it relies more on the IL policy through `AdaConG` to facilitate learning.

### A.4.1 ENVIRONMENT DETAILS

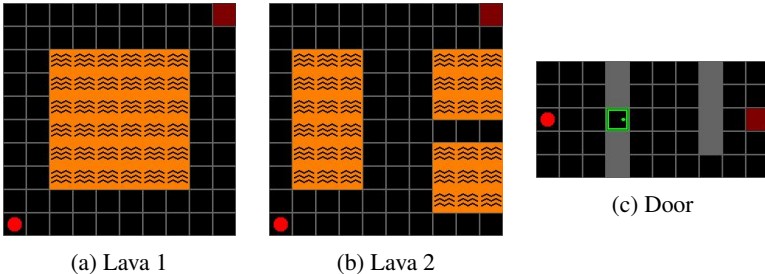

(a) Lava 1       (b) Lava 2       (c) Door

Figure 9: **Gridworld Environment Scenarios.** (a) **Lava 1**: An autonomous agent (red dot) must navigate to a target position (diagonal square) while avoiding lava regions. (b) **Lava 2**: A domain-shifted variant of Lava 1 with altered environment dynamics and layout. (c) **Door**: The agent must traverse a structured environment with doors and walls to reach the designated target position.

### A.4.2 TRAINING DETAILS

For the imitation-guided reinforcement learning experiments, we employ a batch size of 512 and the Adam optimizer (Kingma, 2014) with an initial learning rate of $3e-4$. For each method and environment scenario, we train for 1000 episodes across 10 different random seeds. The model is trained on an Nvidia RTX 3090 GPU with AMD Ryzen 9 5900 CPU and 32 GB RAM.

### A.4.3 EXPERIMENTS WITH ADDITIONAL RL BASELINE

We conduct additional experiments with the RL baseline QDagger (Agarwal et al., 2022) on the Lava 2 environment, which is unseen for the pretrained IL policy. QDagger leverages a teacher policy as a soft imitation regularizer. This mechanism works best when the teacher provides useful corrective guidance on the target environment. However, in our setup, the imitation policy is trained on a different environment and then directly deployed to a new, unseen environment, where it generalizes poorly. As a result, the teacher provides unreliable guidance on the new environment, and QDagger's distillation term ends up regularizing the student toward suboptimal or erroneous actions. QDagger thus underperforms in this setting, whereas AdaConG effectively adapts to the imperfect teacher signals and achieves higher rewards, as shown in Figure 10.

### A.4.4 COMPUTATION OVERHEAD

We compare the training cost of our approach `AdaConG` against key RL baselines, including the pure RL method SAC and IBRL. As shown in Table 12, incorporating `AdaConG` introduces only minimal additional computational overhead.

Table 12: Comparison of computational overhead across key RL baselines.

| Approach | Time/Epoch (s) |
|---|---|
| SAC | 1.87 |
| IBRL | 2.34 |
| AdaConG | 3.08 |

## A.5 AUTONOMOUS DRIVING

### A.5.1 EXPERIMENTAL SETTINGS

We first define the accuracy with respect to a specific degree threshold $\tau$ as $acc_\tau = \text{count}(|\theta - \hat{\theta}| < \tau)/n$, following prior works (Shen et al., 2021; 2023; 2024), where $n$ is the number of test cases;

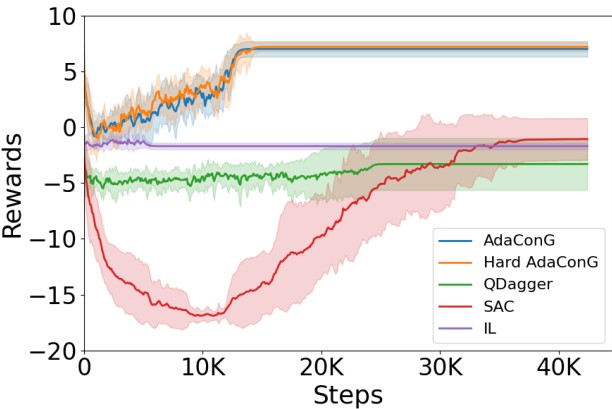

Figure 10: **Additional experiments comparing with the QDagger RL baseline.** QDagger underperforms in this setting whereas AdaConG effectively adapts to the imperfect teacher signals and achieves higher rewards.

$\theta$ and $\hat{\theta}$ represent the ground truth and the predicted steer angle, respectively, for $\tau \in \mathcal{T} = \{1.5, 3.0, 7.5, 15.0\}$. Then we compute the mean accuracy (mAcc) by averaging $acc_\tau$ across different thresholds.

The SullyChen (Chen, 2018) dataset contains approximately 63,000 images, each with a resolution of $455 \times 256$, paired with a corresponding steer angle annotation. We show some sample images in Fig. 11. To generate edge maps from RGB images, we employ DexiNed (Soria et al., 2023). To generate depth maps, we utilize DPT (Ranftl et al., 2021). Following (Shen et al., 2023), we use channel-level attention to represent the importance of each modality. For the teacher model $f_p$, we combine the data from different modalities (RGB, depth, and edge) at the channel level and pass them through an Squeeze-and-Excitation (SE) block (Hu et al., 2018), followed by a $1 \times 1$ convolution layer to make the channel number to be the same as the main modality RGB. We first train the teacher model offline. Then we use it to guide the target student model $f_t$ training through knowledge distillation, while the RGB images for $f_t$ training has domain shifts compared to the ones used for $f_p$ training. For the details of domain shift, please refer to Appendix A.5.2.

We compute nonconformity score $s$ as the residuals between the predicted and true steer angles from the calibration set to get the quantile value $q_{1-\alpha}$ with $\alpha = 0.1$. A sensitivity analysis for different $\alpha$ values is presented in Fig. 12. Then we use it to construct the prediction set $\mathcal{C}(x)$ for a given input RGB image $x$. We define the teacher's uncertainty as the size of the prediction set: $u(x) = |\mathcal{C}(x)|$. The dynamic weight is assigned as $w(x) = 1$ if $u(x) < \tau$, otherwise $w(x) = 0$, for $\tau \in \mathcal{T}$. We set the coefficients $\lambda_{\text{task}}$ and $\lambda_{\text{guide}}$ in Eq. 3 for all knowledge distillation methods following (Shen et al., 2023).

### A.5.2 DATA PROCESSING

After generating depth and edge maps, we split the dataset into $80\%$ training and $20\%$ testing. Then we train the multi-modal teacher model on the training data offline. After training the teacher model, we introduce domain shift to the training data compared to the teacher's pretraining data. Specifically, we add Gaussian noise with zero mean and a standard deviation of 0.1 to $30\%$ of the RGB images, where the noisy samples are selected uniformly at random across the entire training set to ensure consistent noise distribution. We do not add Gaussian noise to the generated depth and edge maps, as they are not used for the student model. Then we shuffle the training data and randomly split it into a $90\%$ training set $\mathcal{D}^t_{\text{train}}$, and a $10\%$ calibration set $\mathcal{D}_{\text{cal}}$, ensuring that both sets are drawn from the same underlying distribution. Additionally, the same Gaussian noise is added to $30\%$ of the testing data of RGB images, to form the noisy test set $\mathcal{D}_{\text{test}}$, which allows us to evaluate the performance of the models under noise conditions.

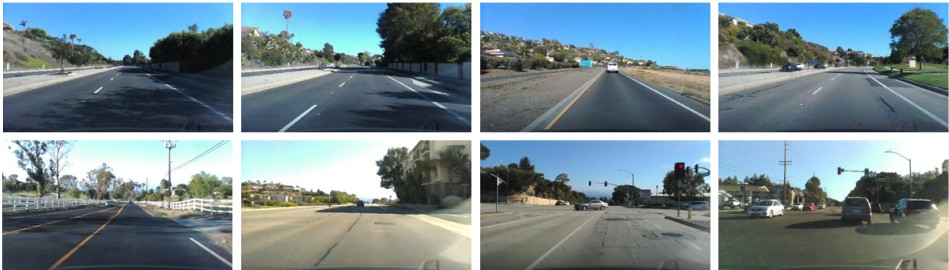

Figure 11: **Sample images of the real-world SullyChen dataset.** SullyChen (Chen, 2018) is a real-world driving dataset which includes diverse driving scenarios with various road types and conditions.

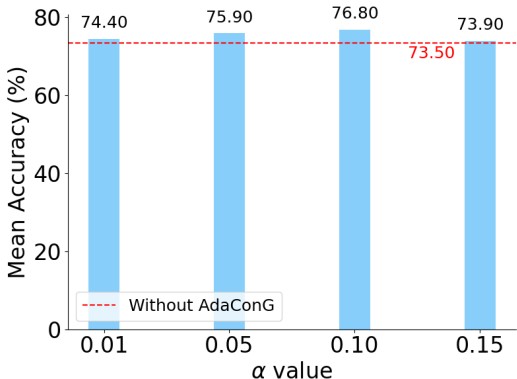

Figure 12: Mean accuracy of KD for steer prediction using `AdaConG` with varying $\alpha$ values.

### A.5.3 HYPERPARAMETER SENSITIVITY ANALYSIS

We conduct a sensitivity analysis for $\alpha$ and present the results in Fig. 12, illustrating how different values of $\alpha$ affect the student model's mean accuracy (mAcc) on the steer prediction task using KD Hinton (2015). We select $\alpha = 0.10$ that yields slightly better performance.

### A.5.4 TRAINING DETAILS

For the experiments, we employ a batch size of 32 and the Adam optimizer (Kingma, 2014) with an initial learning rate of $1e-3$, and a weight decay of $1e-5$. The model is trained on an Nvidia RTX 3090 GPU with AMD Ryzen 9 5900 CPU and 32 GB RAM for 240 epochs.

### A.5.5 COMPUTATION OVERHEAD

We compare the training cost of the key baseline KD and KD with our approach `AdaConG`. As shown in Table 13, leveraging `AdaConG` only introduces minimal computation overhead.

Table 13: Comparison of computational overhead between standard KD and KD with `AdaConG` in autonomous driving steer prediction.

| Approach | Time/Epoch (s) |
|---|---|
| KD | 11.34 |
| KD + `AdaConG` | 13.68 |

## A.6  Discussions and Future Work

`AdaConG` dynamically modulates the influence of guidance signals, allowing models to reduce reliance on potentially misleading information and thereby enhance learning performance. It also suggests promising directions for future work. Currently, `AdaConG` relies on well-defined ground truth; however, its foundation in conformal prediction allows for a natural extension to settings with ambiguous or imprecise labels (Caprio et al., 2025). By designing nonconformity scores that capture label ambiguity, prediction sets can be constructed to reflect varying degrees of uncertainty. This flexibility enables adaptive weighting even under uncertain supervision. Furthermore, integrating concepts from Imprecise Probabilistic Machine Learning, such as credal sets (Caprio et al., 2024) and imprecise probabilities (Dutta et al., 2025), along with strategies from Active Learning and Continual Learning (Lu et al., 2024), could further enhance `AdaConG`'s capacity to manage complex uncertainty.

