# OpenReview forum: "Adaptive Conformal Guidance for Learning under Uncertainty"
_ICLR.cc/2026/Conference — ICLR 2026 Poster_

### Official Review · Reviewer_sHk1 · 2025-10-30

**Soundness:** 3
**Presentation:** 3
**Contribution:** 3
**Rating:** 6
**Confidence:** 2

**Summary:**

This paper proposes AdaConG, that dynamically modulates guidance signals (e.g., from teacher models, pseudo-labels) based on their uncertainty. This uncertainty is quantified using split conformal prediction (CP), which is embedded into the training loop to re-weight the guidance loss. The method is simple, broadly applicable, and validated across diverse tasks (including knowledge distillation, SSL, and imitation-guided RL), showing improved robustness and performance under imperfect guidance.

**Strengths:**

- extensive and diverse experimental setups (also good results)
- simple, easy to re-implement
- minimal computational overhead compared to MC-dropout

**Weaknesses:**

- lack qualitative analysis. For instance, providing visualizations of why AdaConG assigns high or low uncertainty to specific inputs (e.g., showing example images or states and their corresponding CP-derived weights) would offer deeper insight into the method's behavior.
- In RL exps, the comparison is limited to SAC (a standard RL baseline) and IBRL/Soft IBRL (which rely on Q-values). The paper would be strengthened by comparing against other methods that are also designed to handle suboptimal or noisy expert guidance, beyond just Q-value comparison (maybe some baselines in [1]).
- In SSLs, the labeled dataset is used both to construct the calibration set and to compute the training loss. This appears to violate the standard split conformal inprediction assumption. Does it affect the statistical guarantees of CPs?


[1]  Agarwal et al. , Beyond Tabula Rasa: Reincarnating Reinforcement Learning. NeurIPS, 2022.

**Questions:**

None

**Details Of Ethics Concerns:**

None.

---

> ### Author Response · Authors · 2025-11-25
>
> Thanks very much for your comments and feedback. We are glad that you found our paper to have **extensive and diverse experimental setups (also good results), simple, easy to re-implement**, with **minimal computational overhead** compared to MC-dropout. Below we address your concerns and have updated the paper. Hope you could raise the score.
>
> > Qualitative analysis
>
> We add a qualitative analysis demonstrating how AdaConG assigns higher or lower uncertainty to different inputs, providing deeper insight into its behavior. We present the results in Figure 6 in Appendix. We sample multiple clean images from the CIFAR-100 dataset and generate corresponding noisy images by adding Gaussian noise with zero mean and standard deviation 0.05. For both clean and noisy images, we apply a pretrained ResNet-110 model with conformal prediction to obtain the prediction set for each image. We then convert the size of each prediction set into an uncertainty score, then an adaptive weight. The first row in Figure 6 contains the original clean images, while the second row shows their noisy counterparts. For each image, we annotate the prediction set size (S) and the obtained adaptive weight (W). As shown, clean images yield smaller prediction sets and thus higher adaptive weights, whereas noisy images produce larger prediction sets, indicating higher uncertainty and consequently lower adaptive weights for the guidance signal. We also add the qualitative analysis to Appendix A.2.8 in updated paper.
>
> > Additional experiments in RL
>
> We conduct additional experiments with the RL baseline QDagger [1] on the Lava 2 environment, which is unseen for the pretrained IL policy. QDagger leverages a teacher policy as a soft imitation regularizer. This mechanism works best when the teacher provides useful corrective guidance on the target environment. However, in our setup, the imitation policy is trained on a different environment and then directly deployed to a new, unseen environment, where it generalizes poorly. As a result, the teacher provides unreliable guidance on the new environment, and QDagger’s distillation term ends up regularizing the student toward suboptimal or erroneous actions. QDagger thus underperforms in this setting, whereas AdaConG effectively adapts to the imperfect teacher signals and achieves higher rewards, as shown in Figure 10 in Appendix. We also add the additional experiments in RL to Appendix A.4.3 in paper.
>
> > Statistical guarantees of CP in SSL
>
> In the SSL setting,  we do not violate the standard split CP assumption. There is no overlap between the calibration and training sets. The calibration set is constructed from the labeled data, with the same weak augmentation applied as for the unlabeled data. The training data used to generate pseudo-labels (with weak augmentation) come from the unlabeled data. The calibration set is used only within CP to estimate the uncertainty of these generated pseudo-labels, not involved in generating pseudo-labels for the unlabeled set.
>
> [1] Agarwal et al. , Beyond Tabula Rasa: Reincarnating Reinforcement Learning. NeurIPS, 2022.

---

> > ### Comment · Reviewer_sHk1 · 2025-11-27
> >
> > Thanks for the author's response. I will maintain the positive score.

---

### Official Review · Reviewer_YaJZ · 2025-10-30

**Soundness:** 3
**Presentation:** 3
**Contribution:** 3
**Rating:** 8
**Confidence:** 4

**Summary:**

The authors address the problem of noisy and unreliable guidance signals in machine learning. To mitigate this issue, they propose AdaConG, a strategy that reduces the model's reliance on guidance when misleading signals are present and adaptively adjusts the influence of guidance signals. They conduct comprehensive experiments across four popular ML tasks, demonstrating that AdaConG is more robust to guidance uncertainty than baseline methods and generalizes well across diverse tasks and scenarios.

**Strengths:**

1. The experimental design is comprehensive and rigorous, effectively validating the proposed method's robustness to misleading signals.

2. All experimental results report mean and standard deviation over multiple runs, demonstrating scientific rigor and reproducibility.

3. The core idea of improving learning quality by modulating the uncertainty of guidance signals is inspiring, with valuable practical implications across a wide range of tasks.

4. Incorporating conformal prediction to inform training dynamics is novel and represents an underexplored direction that could benefit the community.

**Weaknesses:**

1. I found some of the tables challenging to parse at a first glance (e.g., Table 1 and 4). Reformatting them to be more self-explanatory would enhance the paper's readability.

2. I only found efficiency analysis for KD task. I would recommend the authors also include (training time, computation overhead) comparison in other tasks with key baselines.

**Questions:**

1. What is the baseline for the last section of Table 1? It appears that EA-KD (line 314) outperforms the proposed method (line 301) in half of the cases. Could you clarify this comparison and discuss why EA-KD shows superior performance in these instances?

2. In Fig. 2(d), could you provide the prediction uncertainty curves for other baseline methods? This would help better contextualize the advantages of AdaConG in uncertainty calibration.

3. In Table 4 (last row), under what setting is the result for "Student (RGB)" obtained? Specifically, is this model trained from scratch, or does it use some form of pre-training/initialization?

---

> ### Author Response · Authors · 2025-11-25
>
> Thanks very much for your positive feedback and support. We sincerely appreciate your recognition that incorporating conformal prediction to inform training dynamics is **novel** and **represents an underexplored direction that could benefit the community**, the core idea of improving learning quality by modulating the uncertainty of guidance signals is **inspiring, with valuable practical implications across a wide range of tasks**, the **experimental design is comprehensive and rigorous**, effectively validating the proposed method's robustness to misleading signals, with all experimental results report mean and standard deviation over multiple runs, **demonstrating scientific rigor and reproducibility**. Below we address your questions and have updated the paper.
>
> 1. Thanks for your suggestions. We have reformatted Tables 1 and 4 in paper to make them more self-explanatory. In particular, we now explicitly indicate the teacher models and their accuracies, as well as the student models and their accuracies when trained from scratch.
>
> 2. Thanks for your suggestions. We have included training time comparison in other tasks with key baselines. Please see in the following Tables. Incorporating AdaConG introduces minimal overhead in all tasks. These results are also added in updated paper.
>
> #### Table 1. Comparison of computation overhead between KD, KD + AdaConG, and KD + MC dropout in knowledge distillation.
> |Approach|Time/Epoch (s)|
> |-|-|
> |KD| 6.87|
> |KD + AdaConG|7.04|
> |KD + MC dropout|44.90|
>
>
> #### Table 2. Comparison of computation overhead between FlexMatch and FlexMatch + AdaConG in semi-supervised image classification.
> |Approach|Time/Epoch (s)|
> |-|-|
> |FlexMatch|91.52|
> |FlexMatch + AdaConG|97.63|
>
> #### Table 3. Comparison of computation overhead across key RL baselines.
> |Approach|Time/Epoch (s)|
> |-|-|
> |SAC|1.87|
> |IBRL|2.34|
> |AdaConG|3.08|
>
> #### Table 4. Comparison of computation overhead between KD and KD + AdaConG in autonomous driving steer prediction.
> |Approach|Time/Epoch (s)|
> |-|-|
> |KD|11.34|
> |KD + AdaConG|13.68|
>
> 3. EA-KD is a dedicated KD variant that incorporates an entropy-based mechanism specifically designed for knowledge distillation, and it can yield improvements in certain settings. However, EA-KD is included as a standalone method, rather than a baseline that AdaConG builds upon. Our approach is complementary: AdaConG provides a broadly applicable, model-agnostic uncertainty signal that can be applied across KD, SSL, RL, and other paradigms to improve learning performance, whereas EA-KD is specialized for KD.
>
> 4. We have provided the prediction uncertainty for other baseline methods in RL. Please see the updated Fig. 2(d) in paper. As shown, AdaConG maintains lower prediction uncertainty than other baselines, indicating stronger robustness to uncertainty.
>
> 5. In Table 4 (last row), the "Student (RGB)" model was trained from scratch.

---

> > ### Comment · Reviewer_YaJZ · 2025-11-26
> >
> > Thanks for the authors’ response. My concerns have been adequately addressed. I will keep the positive score.

---

### Official Review · Reviewer_a8X8 · 2025-10-31

**Soundness:** 3
**Presentation:** 2
**Contribution:** 2
**Rating:** 4
**Confidence:** 3

**Summary:**

The paper proposes Adaptive Conformal Guidance (AdaConG), to consider the uncertainty against noisy guidance signals for different tasks. Simply, the method dynamically modulates the influence of guidance signals based on the uncertainty quantified by split conformal prediction (CP), which enabling the models to reduce reliance on potentially misleading signals. The proposed method is evaluated on different tasks, including knowledge distillation, semi-supervised image classification, gridworld navigation, and autonomous driving. The experiments demonstrate the effectiveness of the proposed method

**Strengths:**

1. The paper is well written and easy to follow.

2. The idea of the paper is straightforward.

3. The experiments on diverse tasks demonstrate the effectiveness of the proposed method.

**Weaknesses:**

1. As discussed in the abstract and introduction, the learning-with-guidance methods are easy to be influenced by noisy guidance lead by domain shifts or limited data, where considering the uncertainty is important. However, in the experiments, the datasets and cases are used for evaluation seems more simple, without considering distribution shifts or limited data. I suggest to add some experiments on these difficult cases, such as datasets like CIFAR-C, ImageNet-C.

2. The method is also evaluated on small datasets and backbones. It is not clear how it works on larger backbones like ViT/B, ViT/L, and larger datasets like ImageNet.

3. Lack of comparisons with CP alternatives in the experiments

4. I assume the proposed method would lead to more computational costs for training and calibrating, which are not discussed in the main paper.


larger backbones

larger datasets

**Questions:**

1. The proposed method requires an amount of available target data for calibration during training. However, the target data may not available or hard to collect in some real applications, so the available target data is limited. How could the proposed method handle these cases? How many target data does the method required for calibration and training for different tasks?

2. I'm also curious about whether the proposed method can be extended to self-supervised learning.

---

> ### Author Response · Authors · 2025-11-25
>
> Thanks very much for your comments and feedback. We are glad that you found our paper is **well written and easy to follow**, the **idea is straightforward**, with **experiments on diverse tasks demonstrating the effectiveness** of the proposed method. Below, we address your concerns and have updated the paper. We hope you could raise the score.
>
> > Additional experiments on more challenging dataset CIFAR-100-C
>
> We conduct additional experiments on the more challenging CIFAR-100-C benchmark. We evaluate under the Gaussian noise corruption with five levels of severity, larger severity corresponds to stronger noise and greater distribution shift. More details about the dataset can be found in [1]. We use two pretrained teacher models, ResNet-110 and ResNet-56, both trained on the clean CIFAR-100 dataset, and perform knowledge distillation under different methods, including KD and LS-KD, with both homogeneous and heterogeneous teacher–student structures.
>
> We report the mean and standard deviation over four repeated runs. The results in following table show that incorporating AdaConG consistently improves student performance across all settings. These gains on a more challenging dataset further demonstrate that our method can effectively modulate imperfect guidance signals under distribution shift and lead to improved performance.
>
> We also added these additional experimental results to the updated paper, please see Appendix A.2.6, marked as blue.
>
> |Teacher model|ResNet110|ResNet56|ResNet110|ResNet56|
> |-|-|-|-|-|
> |Teacher accuracy|$24.84$|$20.61$|$24.84$|$20.61$|
> |Student model|ResNet20|ResNet20|ShuffleNet-V1|ShuffleNet-V2|
> |Student accuracy (from scratch)|$41.24\pm0.55$|$41.24\pm0.55$|$35.43\pm0.68$|$35.56\pm0.61$|
> |KD|$24.32\pm0.36$|$20.54\pm0.53$|$24.60\pm0.17$|$20.58\pm0.27$|
> |KD + AdaConG|$42.16\pm0.36$|$39.16\pm0.43$|$35.57\pm0.73$|$37.22\pm0.49$|
> |$\Delta$|$17.84$|$18.62$|$10.97$|$16.64$|
> |LS-KD|$41.02\pm0.52$|$38.58\pm0.08$|$33.93\pm0.70$|$34.94\pm0.92$|
> |LS-KD + AdaConG|$44.44\pm0.38$|$43.33\pm0.21$|$37.76\pm0.33$|$40.34\pm0.63$|
> |$\Delta$|$3.42$|$4.75$|$3.83$|$5.40$|
>
> > Additional experiments with larger backbone and larger dataset
>
> We also conduct additional experiments with a larger backbone and a larger dataset to further validate our approach. Due to the computation constraints and the limited rebuttal period, training with ViT-L and ImageNet is not feasible. Instead, we scale up our evaluation using a larger backbone ViT-B and a larger dataset Tiny ImageNet, which contains 100K training images and 10K testing images across 200 classes. We first train the ViT-B teacher on the clean Tiny ImageNet dataset for 50 epochs. During knowledge distillation to different student models, we introduce noise into the Tiny ImageNet dataset following the procedure described in Appendix A.2.1. This places the pretrained teacher model under distribution shift, making its prediction noisy. In this setting, we leverage AdaConG to dynamically modulate the guidance from the imperfect teacher. The results in the following table show that incorporating AdaConG consistently improves student performance across different architectures, demonstrating that our approach remains effective with larger backbone and larger dataset.
>
> |Teacher model|ViT-B|ViT-B|ViT-B|
> |-|-|-|-|
> |Teacher accuracy|$33.06$|$33.06$| $33.06$ |
> |Student model|ResNet20|ShuffleNet-V1|ShuffleNet-V2|
> |Student accuracy (from scratch)|$43.50\pm0.15$|$50.08\pm0.21$|$51.79\pm0.11$|
> |KD|$31.21\pm0.14$|$32.87\pm0.33$|$33.00\pm0.24$|
> |KD + AdaConG|$43.42\pm0.32$|$51.20\pm0.21$|$52.02\pm0.28$|
> |$\Delta$|$12.21$|$18.33$|$19.02$|
> |LS-KD|$42.16\pm0.39$|$48.59\pm0.16$|$49.76\pm0.17$|
> |LS-KD + AdaConG|$45.50\pm0.28$ | $52.65\pm0.41$ | $54.69\pm0.28$ |
> |$\Delta$|$3.34$|$4.06$|$4.93$|
>
> We also added these additional experimental results to the updated paper, please see Appendix A.2.7.
>
> > Experiments comparison with CP alternatives
>
> Please note that we have already compared CP with several alternatives, including Entropy, MC Dropout, and MSP (maximum softmax probability). The corresponding results are reported in Table 1 of the paper.
>
> [1] Dan Hendrycks and Thomas Dietterich, 2019. Benchmarking neural network robustness to common corruptions and perturbations.

---

> ### Author Response · Authors · 2025-11-25
> **Continuing**
>
> > Computation overhead
>
> We compare the computation overhead of key baselines with and without AdaConG across all tasks, and present the results in the following tables. While incorporating AdaConG introduces a small amount of additional computation, the overhead is minimal in all cases. These results are also added in updated paper.
>
> #### Table 1. Comparison of computation overhead between KD, KD + AdaConG, and KD + MC dropout in knowledge distillation.
> |Approach|Time/Epoch (s)|
> |-|-|
> |KD| 6.87|
> |KD + AdaConG|7.04|
> |KD + MC dropout|44.90|
>
> #### Table 2. Comparison of computation overhead between FlexMatch and FlexMatch + AdaConG in semi-supervised image classification.
> |Approach|Time/Epoch (s)|
> |-|-|
> |FlexMatch|91.52|
> |FlexMatch + AdaConG|97.63|
>
> #### Table 3. Comparison of computation overhead across key RL baselines.
> |Approach|Time/Epoch (s)|
> |-|-|
> |SAC|1.87|
> |IBRL|2.34|
> |AdaConG|3.08|
>
> #### Table 4. Comparison of computation overhead between KD and KD + AdaConG in autonomous driving steer prediction.
> |Approach|Time/Epoch (s)|
> |-|-|
> |KD|11.34|
> |KD + AdaConG|13.68|
>
> > Questions
>
> 1. Yes, AdaConG requires a small amount of target-domain data for calibration. In practice, split CP is very data-efficient: it only needs a small calibration set because it computes a quantile over nonconformity scores, not a full distribution estimate. CP requires only exchangeability and does not rely on sample sizes of the calibration. That said, when calibration data becomes extremely scarce, prediction sets might become large because the quantile estimate is noisier. In such cases, two practical strategies can be applied: 1) Recalibration as more data becomes available. As additional target-domain samples are collected, the calibration set can be incrementally updated to refine the quantile and reduce uncertainty. 2) Adaptive CP with a sliding window. This approach reduces sampling requirements by maintaining a sliding window of recent data and updating quantile online, allowing the calibration to adapt to the evolving policy without requiring large initial datasets.
>
> 2. Yes, AdaConG can be extended to self-supervised learning because CP only requires model scores, not labels. One can conformalize pseudo-targets (e.g., augmentations, contrastive pairs) and adaptively down-weight uncertain guidance, similar to how AdaConG handles pseudo-labels in SSL.

---

### Official Review · Reviewer_rRGf · 2025-11-01

**Soundness:** 3
**Presentation:** 4
**Contribution:** 3
**Rating:** 8
**Confidence:** 4

**Summary:**

The paper proposes Adaptive Conformal Guidance. This is a simple, plug‑in mechanism that uses split conformal prediction (CP) to quantify the uncertainty of guidance signals (teacher logits in KD, pseudo‑labels in SSL, imitation policies in RL) and adaptively modulate how strongly the learner follows them. Concretely, it builds a prediction set (or interval), maps its size/measure to an uncertainty score u, transforms it to a weight and uses it to attenuate a guidance loss (KD/SSL) or to arbitrate between policies (RL). Experiments on CIFAR‑100 KD, SSL (CIFAR‑10/100, STL‑10), MiniGrid navigation, and autonomous driving steer prediction show consistent gains, including large improvements when the teacher underperforms under shift. The paper's appeal in my opinion is breadth and simplicity of the work (of course, it also helps that it is well written and articulated).

**Strengths:**

(1) The idea is elegant, simple and scales well across SSL, KD and imitation RL.

(2) Experiments are rigorous and covers a strong empirical breadth.

(3) It seems to be very lightweight and model agnostic compared to MC-dropout-style approaches.

(4) The paper is well written and articulated.

**Weaknesses:**

Weaknesses and questions:

(1) From my understanding, w(s) weighs a KL guidance loss; in Experiments, however, looks like w(s) chooses actions (stochastic arbitration) and a hard argmax variant. These are different algorithms! Please explicitly state it (if it was not a mistake).

(2) \gamma is being overloaded ((i) RL discount, (ii) temperature in h(u), EMA smoothing factor all use \gamma).

(3) Similarly s is being used for score and state.

(4) Coverage guarantees require exchangeability, several passages imply robustness and in RL the CP set measures self‑consistency of a policy, not correctness. I would rephrase those sentences and state conditions.

(5) In the SAC baseline, generally SAC is meant to be continuous, but how is it using discrete actions?

(6) f_s is undefined (should it instead be \pi_R)?

(7) Typos I was able to catch (there might be more): "Disucssions" (A.6), "mdoel" (Page 9), "is is" (Page 5).

(8) Does the claimed 0.08 ms/sample latency not contradict the epoch timings (\Delta ~= 0.17 s/epoch = ~0.003 ms/sample for 50k samples)?

**Questions:**

Please see the Weaknesses section.

---

> ### Author Response · Authors · 2025-11-25
>
> Thanks very much for your support and positive feedback. We sincerely appreciate your recognition of **the appeal of our work for breadth and simplicity**, our **idea is elegant, simple, and scales well across SSL, KD, and imitation RL**, our **experiments are rigorous and cover a strong empirical breadth**, the method is **very lightweight and model-agnostic** compared to MC-dropout-style approaches, and the paper is **well written and articulated**. Below, we answer your questions and we have updated the paper.
>
> (1) Thanks for your suggestions. We have explicitly stated in paper that the adaptive weight $w(s)$ is also used during data collection: at each state, the agent selects $a_{\mathrm{I}}$ with probability $w(s)$ and $a_{\mathrm{R}}$ with probability $1 - w(s)$. This sampling mechanism is the practical realization of the adaptive guidance and complements the KL-based regularization in the objective. Please see in paper Section 3.
>
> (2-3) Thanks for pointing these out. We have fixed the letters to avoid overloading. We changed temperature in h(u) to $\kappa$, EMA smoothing factor to $\rho$, and score from $s$ to $s'$.
>
> (4)Thanks for the suggestions. We have rephrased the sentences by stating that in RL, the CP set measures a policy’s self-consistency, not action correctness, and we use $w(s)$ as an uncertainty-driven guidance weight. Please see in paper Section 3.
>
> (5) SAC is able to handle discrete actions. We use the discrete variant of SAC, where the policy outputs a categorical distribution over the finite action set rather than a Gaussian. In the experiments of minigrid navigation, we have five actions: left, right, up, down, and stay.
>
> (6-8) Thanks for pointing out these typos. $f_s$ is $\pi_\text{R}$. We have fixed typos such as  "mdoel" and proofread the paper again. 0.08 ms is actually the absolute time cost per sample by inferencing the teacher model with CP on the validation set, not the time added when training with AdaConG. We have fixed all these in updated paper.

---

### Author Response · Authors · 2025-12-03
**Overall AC summary**

Dear Reviewers and ACs,

We sincerely thank all reviewers for their constructive feedback and thoughtful suggestions, which help further strengthen our work. We greatly appreciate the reviewers’ recognition of the following strengths:
- **Breadth and simplicity** (rRGf, a8X8, YaJZ, sHk1)
- **Idea is novel, elegant, straightforward, inspiring** (rRGf, a8X8, YaJZ)
- **Represents an underexplored direction that could benefit the community, with valuable practical implications across a wide range of tasks** (YaJZ)
- **Rigirous, extensive and diverse experimental results** (rRGf, a8X8, YaJZ, sHk1)
- **Lightweight and model agnostic compared to MC-dropout-style approaches** (rRGf, sHk1)
- **Well written and articulated, easy to follow** (rRGf, a8X8)

During the rebuttal period, we conducted additional experiments and provided detailed clarifications to address all reviewers’ concerns. We also updated the paper accordingly. For convenience, we summarize the main updates below:
- **Additional experiments on more challenging dataset CIFAR-100-C**
- **Additional experiments with larger backbone and larger dataset**
- **Training time comparison in all tasks with key baselines**
- **Prediction uncertainty for other baseline methods in RL**
- **Qualitative analysis**
- **Additional experiments in RL with baseline QDagger**
- **Fixed typos and rephrased sentences for clarity**

We would like to sincerely thank all reviewers again for the helpful suggestions that notably enhance the paper.

---

### Meta-Review · Area_Chair_gm9X · 2026-01-04

**Summary:**

This paper presents a general method AdaConG that uses split conformal prediction to quantify the uncertainty of guidance signals and dynamically reduce their influence during training. By embedding conformal uncertainty into the learning loop, AdaConG improves robustness to noisy or misleading guidance across knowledge distillation, semi-supervised learning, and imitation-guided reinforcement learning, with minimal overhead. Reviewers find this proposed idea elegant and novel, "with valuable practical implications across a wide range of tasks".

**Reviewer Concerns:**

Reviewers were generally positive but raised concerns about limited evaluation and presentation of the paper including notation issues and missing details and discussions about assumptions and comparison with additional (non-CP based) baselines. The authors have addressed the main concerns during rebuttal with additional experiments.

**Reviewer Scores:**

During the discussion period, the authors added experiments with a more challenging dataset, larger backbones, and qualitative analysis. The authors also revised the manuscript to address presentation concerns. Two of the reviewers mentioned that they would maintain their positive rating in comment. The authors added results to address most of the concerns of the one reviewer with a negative rating. I believe the reviewer is likely to be willing to raise their score given more time to fully interact.

---

### Decision · Program_Chairs · 2026-01-26

Accept (Poster)